# Plasma-Assisted Nanofabrication: The Potential and Challenges in Atomic Layer Deposition and Etching

**DOI:** 10.3390/nano12193497

**Published:** 2022-10-06

**Authors:** William Chiappim, Benedito Botan Neto, Michaela Shiotani, Júlia Karnopp, Luan Gonçalves, João Pedro Chaves, Argemiro da Silva Sobrinho, Joaquim Pratas Leitão, Mariana Fraga, Rodrigo Pessoa

**Affiliations:** 1Departamento de Física, Laboratório de Plasmas e Aplicações, Faculdade de Engenharia e Ciências, Universidade Estadual Paulista (UNESP), Av. Ariberto Pereira da Cunha, 333-Portal das Colinas, Guaratinguetá 12516-410, SP, Brazil; 2Departamento de Física, Laboratório de Plasmas e Processos, Instituto Tecnológico de Aeronáutica, Praça Marechal Eduardo Gomes 50, São José dos Campos 12228-900, SP, Brazil; 3Departamento de Física, I3N, Universidade de Aveiro, 3810-193 Aveiro, Portugal; 4Escola de Engenharia, Universidade Presbiteriana Mackenzie, São Paulo 01302-907, SP, Brazil

**Keywords:** plasma-enhanced atomic layer deposition, plasma-assisted atomic layer deposition, radical-enhanced atomic layer deposition, plasma-atomic layer etching, ALD, non-thermal plasma, thin-film

## Abstract

The growing need for increasingly miniaturized devices has placed high importance and demands on nanofabrication technologies with high-quality, low temperatures, and low-cost techniques. In the past few years, the development and recent advances in atomic layer deposition (ALD) processes boosted interest in their use in advanced electronic and nano/microelectromechanical systems (NEMS/MEMS) device manufacturing. In this context, non-thermal plasma (NTP) technology has been highlighted because it allowed the ALD technique to expand its process window and the fabrication of several nanomaterials at reduced temperatures, allowing thermosensitive substrates to be covered with good formability and uniformity. In this review article, we comprehensively describe how the NTP changed the ALD universe and expanded it in device fabrication for different applications. We also present an overview of the efforts and developed strategies to gather the NTP and ALD technologies with the consecutive formation of plasma-assisted ALD (PA-ALD) technique, which has been successfully applied in nanofabrication and surface modification. The advantages and limitations currently faced by this technique are presented and discussed. We conclude this review by showing the atomic layer etching (ALE) technique, another development of NTP and ALD junction that has gained more and more attention by allowing significant advancements in plasma-assisted nanofabrication.

## 1. Introduction

Processing based on plasma technology is one of the leading technologies used in the modern world, especially in low-temperature non-thermal plasmas. The term plasma was proposed by Irving Langmuir [1] to describe an ionized gas. In general, substances change phase when energy or heat is added. This shift from solids to liquids (melting) and from liquids to gases (evaporation) occurs at a constant temperature. In contrast, when adding energy to a given gas, neutral particles lose electrons and become ions. Therefore, plasma is an ionized gas and cannot be classified as a phase change from gas to plasma [1]. In a macroscopic view, the plasma has the local ionic density (*n_i_*) equal to the local electron density (*n_e_*), making it electrically neutral. Due to the electron’s tiny mass and its higher mobility, its velocities are more significant than the velocity of other particles. Thus, electrons have a higher temperature than ions and neutral species (see Figure 1). This type of plasma is classified as non-thermal plasma (NTP), where the gas temperature is much lower than the electron temperature at low pressure [2].

Nowadays, the great focus of NTP is on atmospheric pressure applications. This includes uses in air sterilization [4], food preservation and treatment of dormant seeds [5,6], antimicrobial agent in medical and dental areas [7,8,9,10,11,12], and activated liquids to be applied in the fields mentioned above [13,14,15,16]. However, NTP is well consolidated and has been prominent for decades in fabrication and surface modification at low pressure. This highlight of NTP is due to its distinct superiority in the fabrication and modification of materials concerning traditional means of chemical reaction [17]. Figure 2 shows some examples of NTP applications in different areas. Regarding low-pressure plasma application, we can subdivide the main applications resulting from the plasma interactions with the matter as (i) material deposition, (ii) surface etching/modification, and (iii) gas/liquid treatment. In the three subdivisions, the processes are carried out at a low process temperature, which is advantageous compared to conventional thermal processes. This low temperature is only possible because the plasma promotes chemical reactions with electrons that allow it to catalyze reactions that in conventional routes would need extremely high neutral gas temperatures. The high reactivity of plasma-based processes can cause high surface selectivity, greater production efficiency, and a greater variety of starting materials [17,18,19,20,21,22,23]. The preservation of thermosensitive surfaces can also be processed by NTP [2].

Parallel to NTP, more precisely, in the last two decades, the atomic layer deposition (ALD) technique has emerged. This disruptive technology is a specific modification of the chemical vapor deposition (CVD) technique. In just a few years, ALD became the best option to produce ultra-thin films in the order of Angstroms. In CVD, a metallic precursor vapor and a reaction gas undergo dissociation simultaneously under a heated substrate [24]. In contrast, in the ALD technique, the metal reactant and co-reactant are alternately exposed under the heated substrate within an ALD temperature window that prevents thermal decomposition of the metal reactant [25]. Between alternating pulses of reactant and co-reactant, the reactor is purged with an inert gas. Thus, the ALD cycle involves the exposure of the substrate to a metal reactant, cleaning the reactor through a purge, exposure to co-reactant, then another purge [26].

Due the unique characteristics by ALD expand in many fields, as in biomedical engineering [27], corrosion prevention applied in beverage packaging [28,29], textile dry-coloring [30], solar cell technology [31,32], solar water splitting [33], fuel cell [34], membranes and optoelectronics [35,36,37], dental materials [38], pharmaceuticals [39], photosensor and light emission [40], and micro/nanoelectronics [41,42]. Among the unique characteristics stand out the following metrics: the thickness control at the nanometer level, the uniformity of the thin films on large areas, and excellent conformity on irregular surfaces including pores, surface roughness, nanotubes, i.e., 3D patterns in general (see Figure 3) [43]. These metrics stem from the self-limiting nature of ALD, which does not change even with the increased flow of precursors, maintaining the same thickness of thin films regardless of the substrate shape.

It is important to note that all properties and characteristics mentioned above about ALD are fully valid in solid materials. In contrast to solid materials, polymeric materials are less susceptible to self-limiting growth, which is the fundamental characteristic of ALD. Therefore, the great challenge of the ALD process in polymeric materials is the low density of reactive sites on its surface, which considerably alters the nature of the process in these materials. Figure 4a shows the infiltration of molecules from the ALD reactant and co-reactant that enter the subsurface and cause a series of interface reactions, which makes the reactions different from those obtained on solid surfaces [43]. Figure 4b shows the permeability of polymeric materials due to the many polymeric chains and interstitial spaces of different sizes that facilitate the diffusion of small molecules by altering the surface morphology or even altering the internal structure of the polymer [44]. Another point to be highlighted is that due to the various functional groups existing in a polymer (amide, hydroxyl, carboxylic, and their combinations), the formation of intermediate by-products of the reaction occurs, generating new mechanisms of surface reaction that make it complex and difficult to control, being quite different from the ALD process in solids [43,44]. In hydroxyl-rich polymers (high density of reactive sites), the surface reactions mimic those of solid materials with the difference that they produce surface reaction by-products different from standard ALD. In contrast, hydroxyl-poor polymers are more absorbent, so the tiny molecules of the reactant and co-reactant have greater mobility in the substrate, so there is a greater probability of generating subsurface and surface bonds in the polymeric material. Finally, it is essential to highlight that there is a delay in the growth of films by ALD in polymers compared to the process in solid materials. However, the low operating temperature of NTP-assisted ALD is the major attraction for applications in surface modification of natural and synthetic polymers.

Finally, it is essential to mention atomic layer etching (ALE), the most advanced etching technique used in nanofabrication, being considered the counterpart of ALD. ALE is a technique used for the removal of thin layers of material using sequenced and self-limiting reaction steps [45]. The combination of ALE and ALD are crucial for the nanofabrication of processors with sub-5 nm technology. Studies with ALE started more than thirty years ago in research laboratories. Still, as with every new technology, improvements were needed to increase the etch rate (ER), as ALE is a prolonged technique [45]. However, with the addition of NTP, ALE increased the ER by 1000 times [46]. Another critical point is plasma-assisted atomic etch’s potential to smooth and flatten corroded surfaces.

**Figure 3 nanomaterials-12-03497-f003:**
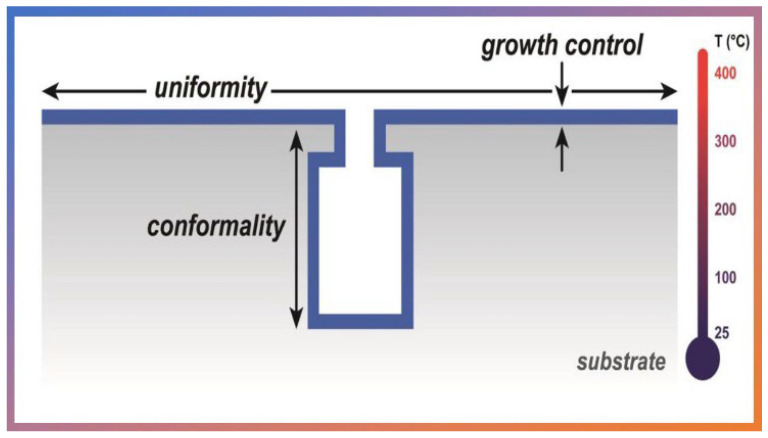
Schematic representation of ALD coverage metrics on 3D substrates. Reprinted with permission from Ref. [47]. Copyright 2014, Elsevier.

Therefore, when we observe the main characteristics of the three techniques: non-thermal plasma, atomic layer deposition, and atomic layer etching, it is unavoidable not to think about combining them to improve and expand ALD/ALE technique. Based on this idea, this article intends to review the junction between NTP and ALD and ALE and show the technological advances based on this junction. To better understand this “marriage” of technologies, Section 2 will discuss the advantages and disadvantages of NTP in ALD. At this point, it will show the types of gases used as plasma sources (co-reactants) in ALD, and we will also show the main types of plasma-assisted ALD reactors (PA-ALD). In Section 3, the main applications of PA-ALD will be presented, outlining the applications in deposition, and surface modification. A new application of NTP technology will be demonstrated to activate ALD precursors to improving reactive species and growth per cycle in ALD. Section 4 will discuss the advantages of NTP in ALE. In Section 5, the main applications of PA-ALE will be presented. Finally, in Section 6, we will discuss the future perspectives of the NTP applied in ALD and ALE.

**Figure 4 nanomaterials-12-03497-f004:**
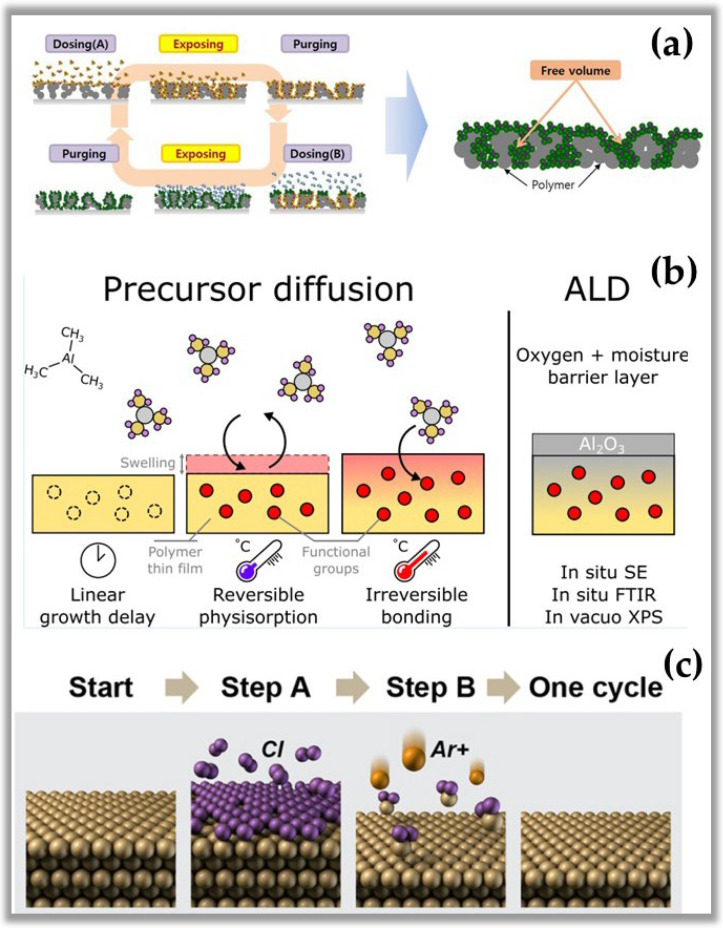
(**a**) Schematic representation of atomic layer infiltration (ALI) [43]. (**b**) Schematic representation of precursor diffusion onto polymer [44]. (**c**) Schematic representation of ALE steps [46]. Subfigure (**a**) reprinted with permission from Ref. [43] by Creative Commons 4.0. Subfigure (**b**) reprinted with permission from Ref. [44] Copyright 2021, ACS. Subfigure (**c**) reprinted with permission from Ref. [46] Copyright 2018, ACS.

## 2. Why Non-Thermal Plasma on ALD?

Commonly in the literature, we find four name variations for the technique that uses NTP in ALD. The most used name is plasma-enhanced atomic layer deposition (PEALD), another variant is plasma-assisted atomic layer deposition (PA-ALD), plasma atomic layer deposition (Plasma ALD), and finally radical-enhanced atomic layer deposition (REALD) [48,49,50]. The first use of NTP in ALD was published in 1991 by De Keijser and Van Opdorp [51]. In this work, they used a microwave-induced plasma for the epitaxial growth of GaAs using H_2_ plasma. However, a gap occurred between the first and second publications, being NTP in ALD again published in 2000 with Rossnagel et al. [52], where metallic Ta and Ti films were deposited. In the last 20 years, the diversification of deposited materials, reactor types, and NTP-enhanced ALD applications has grown dramatically. However, before discussing the types of materials, gases used as a source of plasma generation, and types of reactors, reasons for the successful “marriage” between plasma and ALD will be answered.

First, it is possible to highlight the high reactivity resulting from the reactive species generated in the gas phase that activates the deposition substrates’ surface. The high reactivity generated by the plasma can be adjusted by changing the type of gases, varying the flow and pressure of the gases, changing the plasma power [53,54]. It is also possible to control the bombardment of ions, i.e., the energy dissipated locally, by adding grids between the plasma and the substrate, generating a plasma generation area and another process area, as well as grounding the substrate or changing its polarization [55,56]. All of the adjustments mentioned above help provide a high reactivity supplied to the deposition surface, independent of the substrate’s temperature or the material used as substrate. As seen in Figure 1, NTPs have relatively low heat flux, which is a consequence of T_g_ < T_e_ and the short time intervals of the ALD cycle that prevent the plasma from heating the material. Thus, PA-ALD achieves a higher growth rate per cycle at lower temperatures [57], as shown in Figure 5, reaching thermosensitive materials and areas of technology previously inaccessible by thermal ALD.

This increase in growth per cycle (GPC) at low temperatures is important to the industry, as it reduces film production time and costs. As “time is money,” PA-ALD has become an attractive deposition technique, generating the search of academic and industrial players for new chemical routes and new types of reactors based on NTP.

### Materials, Plasma Gases and Types of Reactors

With the advancement of NTP as an ALD enhancer, the variety of deposited materials has increased considerably, such as oxides [58], nitrides [59], sulfides [60], fluorides [61], phosphates [62], and elemental materials [63]. Table 1 summarizes the main materials deposited by plasma ALD and the main gases applied in plasma generation specifically for the composition of each type of oxide, nitride, sulfides, etc.

A brief search was performed in different databases (Scopus.com and Sciencedirect.com) on 10 August 2022, using the following keywords: * atomic layer deposition, * plasma-enhanced atomic layer deposition, * deposition of a plasma-assisted atomic layer, * ALD plasma, and * radical-enhanced atomic layer deposition. These searches show us that plasma ALD publications represented approximately 10% of the total publications (ALD + ALD plasma) in the last six years. However, in absolute numbers, ALD and plasma ALD publications grow annually. These data indicate that the NTP applied in ALD has a wide development field, mainly due to the recent growth of new deposited materials. Since the first publication by De Keijser and Van Opdorp in 1991 [51], when they used a microwave-induced plasma for the deposition of GaAs, to the present day, there has been a significant increase in the types of reactors. Some of these reactors are commercial, and others are for research use. The reactors can be divided into three main groups: (1) direct plasma; where the substrate is within the plasma generation zone, (2) remote plasma; where the substrate is located outside the plasma generation zone and (3) radical-enhanced; where the substrate is reached only by radicals with sufficient useful life. In the case of direct plasma, the idea is to obtain a high flux of species with high activation energy. In contrast, the remote plasma is configured to limit or control the flow of highly energetic ions and photons, thus preventing damage or corrosion of the film or substrate [55]. Figure 6 and Table 2 summarize the reactor subgroups of each type and show their schematic configurations.

In this section, a wide variety of reactor solutions existing in plasma ALD designs were presented. Plasma generated by microwave surfatron, capacitively coupled plasma (CCP), inductively coupled plasma (ICP), hollow-cathode (HC), microwave electron cyclotron resonance (ECR), and dielectric barrier discharge (DBD) are not new technologies in the field of deposition. However, they are new technologies incorporated in ALD. This evolution in plasma ALD reactors is due to physicists, chemists, engineers, materials scientists, and industrial players who have studied and selected the right tools crucially in the advancement and high performance of NTP applied in ALD. In all of the reactors presented, the control of processing parameters, such as ionic flux and pressure, are the main factors responsible for fine-tuning the structural properties of nanomaterials grown using the plasma ALD technique.

## 3. Main Applications of PA-ALD in Nanofabrication

This section will review the last papers on plasma-based technology applied to enhance ALD. Firstly, the main applications of different types of thin films grown by ALD plasma (oxides, nitrides, sulfides, fluorides, phosphates, and elemental materials) will be discussed. The second point to be addressed is the surface modification caused by plasma ALD on some surfaces, such as polymers. The third point to be addressed is 2D materials, which play a crucial role in micro/nanoelectronics due belong to a class of advanced materials with odd properties. It is worth mentioning that in Appendix A, there is a summary of our group’s main contributions to the area.

### 3.1. Deposition: Plasma-Assisted Atomic Layer Deposition

In this section, we will briefly review the main applications of thin films grown by plasma ALD. The focus of this section is to show the versatility of the plasma ALD technique in the deposition of different materials using different reactors, precursors, and present old and new applications.

#### 3.1.1. Oxides

ALD is one of the most used techniques in photovoltaic (PV) technology, mainly for buffer layer growth (BFL) and passivation layers (PSL). These layers generally increase the rate of recombination by reducing interface defects, which increases power conversion efficiency (PCE). It is important to highlight that PV technology encompasses all existing technologies, such as silicon wafer-based PV [68], chalcogenide thin-film-based PV [69], III-V based PV [70], perovskite-based PV [71], dye-sensitized based PV [72], organic-based PV [73], and quantum dot solar cells [74]. However, new applications that go beyond BFL and PCE are being studied.

In 2016, Wang et al. [75] showed that tin oxide (SnO_2_) deposited at low temperatures by PA-ALD has an excellent electron selective effect in highly efficient organic-inorganic metal halide perovskite solar cells with a planar cell structure. Figure 7 shows the PCE as a function of the ALD cycle and temperature process, and as can be seen, the thickness that optimizes the PCE is 17 nm for SnO_2_.

Tetrakis(dimethylamino)-tin(IV) was used as the Sn reactant, heating the precursor line to 75 °C. Pure O_2_ was used as the co-reactant, and ultra-pure argon (Ar) was used as the carrier gas with a flow rate of 15 sccm. The resulting growth per cycle (GPC) was about 0.17 nm/cycle. The importance of PA-ALD is due to the temperature reduction of the complex planar cell construction process, which reduces the associated costs. The electron selective layer (ESL) of SnO_2_ was deposited at 70 °C (chamber and substrate temperature), and a PCE of 19.03% for glass substrate and 16.80% for flexible polymer substrate was achieved. At this temperature, the SnO_2_ films that had grown were of excellent quality. This high quality at low deposition temperature enables the large-scale manufacture of efficient perovskite solar cells. In 2018, Reichel et al. [76] studied the PA-ALD of aluminum oxide (Al_2_O_3_) applied as surface passivation and electrical insulation on silicon solar cells. They carried out the experiments in a PA-ALD tool with a remote ICP at 140 W. Trimethylaluminum (TMA), and O_2_ plasma was used as a reactant and co-reactant, respectively. Argon (Ar) purge gas flow 30 sccm was applied. The deposition temperature varied from 100 to 350 °C, with a GPC decrease of 0.16 to 0.1 nm/cycles for the highest temperature. ALD cycles were chosen to generate thin films with thicknesses of about 10 nm to almost 250 nm. They studied the pinhole density and the stress of the thin films as deposited and after annealing from 250 to 450 °C. Despite Al_2_O_3_ thin films being much explored as PSL, the authors’ approach is interesting because the annealing step at 450 °C to films deposited at 250 °C with thickness at least 80 nm presented a low defect density and low stress avoiding a leakage current and generating a high breakdown voltage. These parameters make the Al_2_O_3_ thin films grown by PA-ALD a promising candidate for silicon solar cells that rely on passivating and insulating thin films. In 2020, Wang et al. [77] showed interesting applications of PA-ALD in perovskite solar cells. In contrast to the increase in PCE, perovskite technology suffers from low stability and high toxicity [78]. These questions are the main challenges in the commercialization of perovskite solar cells, despite the constant improvement in the long-term stability of perovskite solar cells, including changes in interface physics, compositional changes, and passivation in the grain boundary. Perovskite quickly degrades in contact with moisture and despite being a promising technology, it barely can support practical applications. In this context, the authors used a new efficient device encapsulation technique to prevent moisture degradation and prolong operational life in a natural environment. The method uses PA-ALD to create an anti-water encapsulation deposited on the perovskite solar cells. Ethyl glycol was first deposited as BFL by molecular layer deposition (MLD), followed by Al_2_O_3_ PA-ALD deposited at 50 °C to prevent possible degradation of metal halide perovskite solar cells. TMA and O_2_ plasma were the reactants of Al_2_O_3_, with Ar as the purge gas. The plasma power was fixed at 100 W. The ALD pulses were 0.04–80–10–120 s, respectively, TMA-Purge-O_2_ plasma-Purge. The authors showed that the encapsulation structure exhibited a water vapor transmittance rate (WVTR) of 1.3 × 10^−5^ g.m^2^⋅day^−1^, the lowest value among thin-film encapsulation layers reported in the literature for solar cells. They also demonstrated that the perovskite solar cells withstood 80% relative humidity (RH) at a temperature of 30 °C for over 2000 h with a 96% preservation of their initial PCE (see Figure 8).

Organic light-emitting diodes (OLEDs) are already a reality in flexible displays and displays marketed worldwide. Industrial and academic interest in research with OLEDs is growing due to their attractive requirements, which are [79]:Wide viewing angle.Low latency.Low power consumption.Ultra-thin lightness and thickness.Mechanically flexible.

However, vulnerabilities caused by contact with moisture and oxygen are still significant barriers. OLEDs are low-work function metal-based technologies that work with active layers of small molecules that are easily oxidized, crystallized, and delaminated after rapid exposure to oxygen and water. This fragility causes edge shrinkage and can also introduce dark spots [80]. As observed in the perovskite solar cells, the most viable solution is the efficient nanoencapsulation of the OLEDs to prevent the penetration of water or oxygen, thus avoiding the degradation of the organic molecules. In this sense, PA-ALD allows for the controlled self-limited growth of high-density, with ultra-thin films conformal that grow at low temperatures by monolayer-by-monolayer with WVRT of about 1 × 10^−6^ g.m^2^⋅day^−1^, which is the necessary value for a perfect operation of OLEDs.

In 2016, Hoffmann et al. [81] were the first to publish a study of thin-film gas permeation barriers based on Al_2_O_3_ cultured by atmospheric pressure plasma-enhanced atomic layer deposition (APPALD). The work was not applied directly to OLEDs but opened the way for new applications. The authors grew Al_2_O_3_ thin films using TMA, and Ar/O_2_ plasma at 80 °C. The substrates used for the analysis were polyethylene terephthalate coated with tin oxide and indium, another substrate used was Si. It is worth mentioning that these films were grown at atmospheric pressure and showed a WVRT of the order of 5 × 10^−5^ g.m^2^⋅day^−1^. In the same year, Kim et al. [82] examined the effects of the oxide ratio on the WVTR of Al_2_O_3_/TiO_2_ nanolaminate films (50 nm) prepared by PA-ALD. They used TMA as a precursor of Al_2_O_3_ and tetrakis-(dimethylamino)-titanium (TDMAT) as the precursor of TiO_2_. O_2_ plasma was used as an oxygen source for both precursors at 100 W. The authors demonstrated that the Al_2_O_3_/TiO_2_ nanolaminate film exhibited optimal properties for a 1:1 atomic ratio of Al_2_O_3_/TiO_2_ with the lowest value of WVTR of 9.16 × 10^−5^ g.m^2^⋅day^−1^ at 60 °C and 90% RH (shelf-life can be observed in Figure 9).

OLED devices showed a most extended shelf-life, over 2000 h, without forming dark spots or edge shrinkage. In 2020, Jin et al. [83] studied the encapsulation of CsPbBr3 (perovskite) nanocrystals (NCs) using thin films of SiO_2_ grown by PA-ALD and showed that the optical performance of the device was minimally affected. The authors showed that the ALD layer effectively prevented the deformation and sintering of CsPbBr3 NCs, improving water, light, and heat stability. They proved that white LEDs encapsulated extended the color gamut to 126% NTSC, suggesting potential application as wide color gamut LED backlights. Therefore, it is clear that the encapsulating used in OLEDs and LEDs by PA-ALD shows promising results and has great industrial appeal.

The most traditional application of oxides grown by PA-ALD is in micro/nanoelectronics. The most conventional devices that take advantage of PA-ALD are transistors, capacitors, and non-volatile memories. Below is a summary of the main applications of PA-ALD:(1)Yeom et al. [84] grew indium oxide (InOx) using Et_2_InN(SiMe_3_)_2_ and O_2_ plasma as reactants. The deposition temperature varied between 100 and 250 °C with a GPC of 0.145 nm/cycle. The authors showed that thin-film transistors (TFTs) built with nano-crystalline InO_x_ had a higher carrier density as the temperature ramp increased from 150 to 250 °C. This effect is due to oxygen deficiency at higher deposition temperatures. The TFTs showed high linear mobility of 39.2 cm^2^ V^−1^ s^−1^ for nano-crystalline InOx grown at 250 °C. Therefore, this result indicates that InO_x_ TFTs are a strong candidate for next-generation high-performance TFTs.(2)Egorov et al. [85] used PA-ALD for TaO_x_ deposition with a controllable concentration of oxygen vacancies (V_O_). Ta(OC_2_H_5_)_5_ was used as a reactant of Ta, and Ar/H_2_ plasma as a co-reactant. The V_O_ control made through the fractional mixture of Ar and H_2_ in the plasma generation was responsible for controlling the leakage current of the resistive switching memory devices in the range of five orders of magnitude compared to the Ta_2_O_5_ film grown via thermal ALD. Ultimately they used the Ta_2_O_5_/TaO_x_ stack with reliable resistance switching up to approximately 10^6^ switching cycles, in contrast to the single-layer Ta_2_O_5_ memory, which demonstrated an order of a few hundred switching cycles.(3)Jha et al. [86] grew HfO_2_ thin films by PA-ALD for non-volatile memory applications. For this, they investigated the electrical and ferroelectric properties of metal-insulator-semiconductor (MIS) and metal-insulator-metal (MIM) capacitors with different thicknesses between 5 and 20 nm deposited on Si and TiN/Si. The best results were achieved for 10 nm films. A memory window of 4 V was found for the MIS structure and the MIM structure, with a maximum remaining polarization of 4 μC/cm^2^. The non-volatile memories based on HfO_2_ thin films showed a data retention structure of more than ten years with a fatigue resistance of 10^12^ read/write cycles.(4)Henning et al. [87] used an aluminum oxide monolayer (~0.3 nm) deposited by PA-ALD for gallium nitride encapsulation in the c-plane (GaN), which is enabled by the partial conversion of surface oxide GaN to AlO_x_ using sequential exposure to trimethylaluminum (TMA) and H_2_ plasma. This thin AlO_x_ monolayer decreases the work function that increases the reactivity with phosphonic acids under standard conditions, leading to self-assembled monolayers with densities close to the theoretical limit. This high reactivity of TMA with surface oxides opens the opportunity to extend this ultra-fine (<1 nm) aluminum oxide deposition approach to other III-V-based dielectrics and semiconductors, with relevance for applications in (photo) electrocatalysis, optoelectronics, and chemical sensing.(5)Xiao et al. [88] studied the effect of HfO_2_ and ZrO_2_ on n-channel p-channel metal oxide semiconductor field-effect transistors (nFETs and pFETs), CMOS inverters, and CMOS ring oscillators were fabricated to test the quality of HfO_2_ thin films, and ZrO_2_ applied as a gate oxide. They used tetrakis(dimethylamino), hafnium (Hf[N(CH_3_)_2_]_4_), and tetrakis (dimethylamino) zirconium (IV) (Zr [N(CH_3_)_2_]_4_) as precursors, respectively, of Hf and Zr. The manufactured nFTTs and pFETs have good electrical properties of n or p-type field-effect transistors, while CMOS inverters based on HfO_2_ and ZrO_2_ have good electrical transfer characteristics. Both manufactured ring oscillators demonstrated satisfactory oscillation waveforms, and the ZrO_2_ gate oxide caused the oscillator to oscillate faster than the HfO_2_ gate oxide.

#### 3.1.2. Nitrides

Like oxides, nitrides are also extensively studied and applied in a range of fields of modern technology, ranging from dielectric barriers to corrosion-resistant films. In this section, we will cover some applications of nitrides grown by PA-ALD. Initially, we will address applications of Al, Ti, and Si-based nitrides.

For example, the application performed by Otto et al. [89] in 2016, where PA-ALD was used to deposit TiN films. They showed the possibility of using it as a promising synthetic plasmonic metal. TiN thin films are an alternative to the plasmonic community with an immediate need to supply traditional plasmonic materials, such as Ag and Au, which have low mechanical, thermal, and chemical stability. Synthetic metal and plasmonic alloys have better stability, but they require growth temperatures > 400 °C, making the use of technological substrates that are generally sensitive to high temperatures unfeasible. In this sense, the work presented by Otto and co-workers using PA-ALD at low temperatures is very important for the area in question. This work used tetrakis(dimethylamino)titanium (TDMAT) as Ti precursor, and N_2_/H_2_ plasma was used as N precursor. Through optical measurements and theoretical modeling, it was demonstrated that the plasmonic network performance had a period of 900 nm. In 2018, Krylov et al. [90] deposited TiN films by PA-ALD and obtained films with low resistivity. According to studies, electron scattering on grain boundaries is the dominant mechanism. This mechanism is a determinant for the low resistivity of TiN films, and it is necessary to obtain large grains in the columnar form to reduce resistivity further. These two examples are essential to see how the same technology and films can be applied in different fields. Kim et al. [91] used PA-ALD to grow SiN_x_ thin films used as 10 nm thick dielectric barriers to prevent the diffusion of Cu into SiO_2_/barrier/Cu/TaN structures on Si substrates. A direct plasma ALD process was used with trimethylsilane (TMS) as the Si precursor, NH_3_ plasma, and He as the purge gas. Although PA-ALD SiN_x_ film has a low dielectric constant (<5), the layer’s ability to prevent Cu diffusion was equivalent to that of plasma-enhanced chemical vapor deposition (PECVD) SiCN. This result shows the feasibility of immediately replacing traditional PECVD-grown SiCN films with PA-ALD SiN_x_ in thin dielectric barrier devices for future advanced technologies. Seppänen et al. [59] used a new approach to grow aluminum nitride (AlN). First, the film is developed by PA-ALD with a plasma power of 100 and 200 W. After film growth, in situ atomic layer annealing (ALA) is used. This possibility of growing crystalline films using ALA to form better quality nucleation layers for further growth of nitride compounds is an essential approach for applications in the semiconductor industry.

Han et al. [92] used a novel complex precursor bis((2 (dimethylamino)ethyl)(methyl)amino)methyl(tertbutylimido)tantalum for growth of tantalum nitride (TaN) films. PA-ALD used NH_3_ plasma, and the growth temperature varied between 150 and 250 °C with 0.062 nm GPC. They showed that ultrathin TaN films (2 nm) have a film density of 9.1–10.3 g/cm^3^ at 200–250 °C. Despite only 2 nm, the Cu barrier performance of the TaN film showed excellent properties when evaluated by annealing between 400 and 800 °C. The properties of TaN are comparable to SiN_x_ (10 nm) presented by Kim et al. [91].

Wang et al. [93] developed zirconium nitride (ZrN) + zirconium oxide (ZrO_2_) alloys by PA-ALD to be applied as thin films with high corrosion resistance. The new zirconium oxynitride (Zr_2_N_2_O) coating was deposited on 304 stainless steel (SS) by incorporating a controlled amount of oxygen into ZrN using PA-ALD. The main idea of this work is to solve a double problem involving the susceptibility to corrosion of metallic bipolar plates that limits their application in polymer electrolyte membrane fuel cells (PEMFCs). The candidate material to solve this problem must be conductive and have high corrosion resistance. In this context, zirconium nitride (ZrN) exhibits increased corrosion resistance but lacks conductivity. In contrast, ZrO_2_ is a good conductor but has low corrosion resistance. Therefore, the solution adopted by the authors to build Zr_2_N_2_O thin films proved to be very promising, as the corrosion current density of the specimen coated with Zr_2_N_2_O has an order of magnitude smaller than that of the substrate coated with ZrN. And in long-term testing, the interfacial contact resistance (ICR) of the Zr_2_N_2_O coated specimen is much lower than that of the ZrN coated sample due to improved oxidation resistance. As shown in Figure 10, the potentiodynamic polarization curves show that the adopted strategy produces thin films with an ultra-high corrosion resistance with considerable conductivity.

Sowa et al. [94] and Tian et al. [95] grew thin films of niobium nitride (NbN) that have excellent physical, chemical, and electrical properties. NbN PA-ALD were deposited with (tert-butylimido)-tris(diethylamino)niobium as the Nb reactant and plasma H_2_/N_2_ and H_2_/NH_3_ as the N source. Sowa et al. used temperatures between 100 and 300 °C and powers ranging between 150 and 300 W. At high deposition temperatures, the 350 nm thick films had more significant cubic NbN crystals and had higher density with resistivity at room temperature of the order of 173 μΩ.cm and with superconductivity analysis of 13.7 K. Tian et al. used temperatures between 200 and 400 °C and plasma power fixed at 2800 W. The annealing of the films was used (50 µm thick) to improve the crystallinity and was carried out in an Ar environment with temperatures ranging between 800 and 1200 °C and exposure times varying from 10 to 60 min. The efficiency of superconducting radiofrequency cavities composed of Nb improved after thermal annealing up to 13.8 K, a value compatible with the desired application. An increase in density was also observed, as well as an increase in grain size.

#### 3.1.3. Sulfides, Phosphates and Others

Sulfide-based thin films started to stand out, mainly due to the technological appeal of materials such as tungsten disulfide (WS_2_) and molybdenum disulfide (MoS_2_), which can revolutionize nanoelectronics [96].

Transition metal-based 2D dichalcogenide, more precisely, WS_2_, is a potential low-dissipation semiconductor material for nanoelectronic devices. However, applications of these materials in nanoelectronics require the materials to grow in crystalline form with the number of monolayers controlled and at low temperatures. In this sense, PA-ALD meets these requirements. Groven et al. [97] grew strong textured nanocrystalline WS_2_ at 300 °C. The PA-ALD reaction cycle consisted of a WF_6_ reaction, an H_2_ plasma reaction, and an H_2_S reaction. They reported that the H_2_ plasma was essential as it reduced surface species −W_6_ + F_x_. In 2019, Balasubramanyam et al. [98], intending to apply WS_2_ as electrocatalysts for the sustainable production of H_2_ through the electrochemical reaction of hydrogen evolution (HER), improved the quality of the deposited films by mixing H_2_+H_2_S as plasma-generating gas in PA-ALD. This approach adopted by the authors helped in satisfactory control of the edge location density and increased the HER performance of the edge-enriched WS_2_ electrocatalyst.

Recently, Sharma et al. [99] and Vandalon et al. [100] synthesized MoS_2_ and Al-doped MoS_2_ films at low temperatures, respectively. These extrinsically doped 2D and 2D semiconductors are essential for the fabrication of high-performance nanoelectronics. Sharma et al. used PA-ALD to grow thick MoS_2_ films from monolayer to multilayer at low temperatures. In contrast, Vandalon et al. synthesized Al-doped MoS_2_ thin films, resulting in a particularly sought-after p-type 2D material. They showed precise control over carrier concentration in the range of 10^17^ to 10^21^ cm^−3^. In the article by Vandalon and co-workers, it is evident that fine control over doping concentration, combined with compliance and uniformity and the sub-nm thickness control inherent in PA-ALD, ensures compatibility with the large-scale manufacture of this material technological.

Cadmium sulfide (CdS) based buffer layers are well established in thin-film photovoltaic technologies, particularly in Cu(In,Ga)Se_2_ (CIGS), CuZnSn(S,Se)_4_ (CZTSSe), and CuZnSnS_4_ (CZTS) type solar cells. However, the new environmental regulations suggested by the UN and proposed by the European Union (EU) [32] indicate the development of solar cells without toxic materials, such as Cd. CdS have other disadvantages that accelerate the change process, ranging from losing absorbed light due to the high recombination of minority carriers to decreased PCE. In this sense, new BF layers need to be presented and studied. Bugot et al. [101] proposed the growth of ultrathin films of In_x_S_y_ and In_2_(S,O)_3_ grown by PA-ALD, which can be implemented as ultrathin interfacial buffer layers in CIGS solar cells. Films were produced using indium acetylacetonate(In(acac)_3_), hydrogen sulfide, and an Ar/O_2_ plasma as indium, sulfur, and oxygen precursors. The authors showed a study that led to a readjustment of the deposition conditions of In2(O,S)_3_ thin films allowing the promising implementation as buffer layers in CIGS solar cells with high PCE.

The versatility of PA-ALD allows the constant creation of new alloys or deposition of metals on 3D structures. This behavior allows for continuous evolution in nanoelectronics and spintronics. Wang et al. [102] grew Ti_0.28_Sb_2_Te_3_ (TST) based films by PA-ALD that can be used as a storage class memory phase change (SCM) material. They showed that due to the fast crystallization and the low melting temperature of the TST, the response speed decreased to 6 ns, while the reset voltage could be reduced by 20% compared to the Ge_2_Sb_2_Te_5_ (GST) based device with the same cell structure. These results indicate that TST thin films synthesized with PA-ALD are a fast and scalable phase change material that can be applied to SCM.

Giordano et al. [103] used PA-ALD to prepare thin films of formed nickel and nanotubes using nickelocene as a reactant, water as a co-reactant agent, and a plasma-enhanced reduction step cycled with H_2_, which is crucial for single material deposition. They fabricated several micrometer-long nickel nanotubes with diameters ranging from 120 to 330 nm (Shown in Figure 11). Their results show that the spin-wave damping was low, allowing detection of several stationary spin-wave modes, which fulfilled the condition of constructive interference in the azimuthal direction of a nanotube. Therefore, Ni planar films and nanotubes exhibited physical properties promising for functional spintronic elements and magnetic applications in 3D device architectures.

### 3.2. Modification: Surface Modification on Polymers

NTP technology is widely used in the surface modification of polymers to bring the desired functionality to the polymer surface. However, the inhomogeneity and hydrophobicity of some polymers are incompatible with other substances necessary for the further functionalization of the polymer. DBD and gliding arc (GA)-based plasma generator reactors are commonly used in an attempt to overcome these problems that reduce functionalization [104,105]. The modification generated by these types of plasma can lead to a chemically unstable and non-uniform surface, which results in a low to moderate performance in the final application [106]. In this context, PA-ALD at low temperatures has low reaction rates, which result in low deposition rates. These unique features added to the ability to deposit ultra-thin films, making PA-ALD an essential tool in the surface modification of polymers, being a helpful way to bring the desired functionality to the polymer surface. However, some technical problems need to be overcome. Unlike hard inorganic surfaces that are commonly used as substrates in PA-ALD and have many reactive sites that promote film growth, reactive sites may or may not be present in the case of polymers. There is small molecule permeability in many polymers, allowing for the diffusion of PA-ALD precursors into and out of the polymer substrate subsurface [106]. Thus, we can classify two polymer types: those that imitate a solid surface and have several reactive sites for film growth and the others that result in different final products. It is noteworthy that although reactive sites facilitate chemisorption between gas molecules and reactive functional groups on the surface, this does not always lead to smooth conformal growth as expected for PA-ALD on a solid and inorganic substrate [107]. In most polymers, the precursor diffuses to the sub-surface, causing rough surface texture, polymer swelling, and particle growth below the surface [108]. Therefore, this section will discuss recent studies of polymer functionalization by PA-ALD, showing that the surface reactions will depend on the nature of the polymer, the chemistry of the precursor and co-reactant, and the parameters of the plasma used in the process.

Semi-crystalline polymeric substrates such as polyethylene terephthalate (PET) and polyethylene naphthalate (PEN), and amorphous substrates such as polyethersulfone (PES), due to their transparency and flexibility, are gaining attention as promising materials applied to flexible devices. PET coated with transparent conductive oxides is applicable in OLED devices [109]. PEN coated with carbon nanotubes is used as flexible solar cells, OLEDs, and touch panels [110]. PES have high thermal resistance for operation at high temperatures, so they have great appeal in flexible optoelectronics [111]. However, these semi-crystalline or amorphous polymers have many technical problems to be solved. In particular, moisture and oxygen permeation damage substrates, reducing the life of OLEDs, flexible solar cells, and touch panels. The cited problem is avoided or mitigated by the use of permeation barriers which are usually oxides. In 2017, Kim et al. [108] and Fang et al. [112] used low-temperature PA-ALD to study the barrier properties of fine oxides. Kim and co-authors showed that the WVTR of the Al_2_O_3_ layer deposited on the plasma-treated PEN substrate was about 7.2 × 10^−4^ g.m^2^⋅day^−1^, which was significantly lower than that of the untreated substrate. Prior to PA-ALD, they pre-treated the polymer substrates with argon and oxygen plasmas to increase the number of reactive sites and facilitate the formation of polar groups that are crucial in the PA-ALD process. Fang et al. significantly reduced the WVTR to less than 10^−3^ g.m^2^⋅day^−1^ without a long-term cooling process, for PET substrates. A similar study was carried out by Lee et al. [113], but the material chosen was SiO_2_ grown by PA-ALD. They reduced the WVTR to 7.73 × 10^−5^ g.m^2^⋅day^−1^, which was 10 times lower than a control sample. Therefore, the growth of oxides by PA-ALD significantly enhances the protective barrier on flexible devices.

Kovacs et al. [114] presented in situ gas permeation measurements to study the cracking of oxide films grown on low-density polyethylene. They investigated the application of nanoscopic defects in Al_2_O_3_ thin films deposited by PA-ALD in polymeric electrolyte fuel cells. They showed that this mechanical limitation, which can impair their encapsulation utility, is critical in the case of polymeric electrolytic fuel cells containing a proton exchange membrane, where water retention in the membrane is crucial for the efficient transport of hydrated ions. Kovács and co-workers demonstrated that these nano-cracks could act as nano valves at low humidity levels and maintain the proper humidity level for the membrane.

In 2021, Sun et al. [115] proposed a new strategy for designing and manufacturing new integrated portable electronics. In contrast to previously reported studies that focus on encapsulation of devices. Sun and co-workers present a promising way to manufacture integrated, flexible, multifunctional sensors. Due to toughness and environmental stability, they used poly(vinylidene fluoride) (PVDF) as a polymer substrate to manufacture flexible strain sensors. The PVDF was pretreated with Ar plasma facilitating the deposition of the piezoresistive ZnO nanolayer via PA-ALD. ZnO/PVDF-based strain sensors exhibited high performance to monitor strain below 6%, especially in ranging from 0.1% to 0.6%, which are extremely difficult to detect by other methods accurately. Indeed, this article opens up new options for applying PA-ALD over polymers and expands the construction options for wearable sensors to be designed.

### 3.3. Efforts to Produce High-Quality 2D Layered Structures

Since the discovery of graphene, there has been a growing race to produce two-dimensional (2D) layered materials. This continuous search for 2D materials is due to the exceptional properties these materials have. ALD and, consequently, PA-ALD show promise in constructing these materials, mainly due to the unique growth mechanism at the nanometer scale.

In this sense, three distinct methods of ALD in developing 2D materials stand out. Figure 12 illustrates these three methods according to Cai et al. [116]. The first method is based on the conventional ALD process (Figure 12a) and is still the most used method for the processing of amorphous, polycrystalline, or hybrid (amorphous/polycrystalline) 2D materials [117,118]. The net adsorption force between the gaseous reactants and the substrate surface is used. This net adsorption force is always positive with increasing film thickness, allowing for layer-by-layer growth typical of ALD. However, this conventional method allows for slow lateral growth. The second method (Figure 12b) is based on the temperature-focused self-bounding layer synthesis process to define the number of layers [119,120]. Unlike conventional ALD, which is based on the number of cycles to determine the number of layers. However, this method is rarely used and is limited to the growth of MoS2 [119] and WSe2 [120]. The advantages of this method lie in (i) the possibility of controlling the layer number precisely, (ii) uniformity on a large scale, and (iii) the production of 2D materials with a high degree of crystallinity. The disadvantages are that (i) the net adsorption on the substrate surface decreases with an increasing number of layers, becoming zero at a saturated number of atomic layers, and (ii) the method requires high temperatures (>generally above 500 °C) to achieve self-limited growth of crystalline 2D materials [116,119,120]. The third method (Figure 12c) is based on two process steps. First, a film is grown at low ALD processing temperatures, then post-treatment is performed at high temperatures to convert the material into 2D. This method has the advantage of manufacturing several 2D materials, such as MoS_2_, MoSe_2_, WS_2_, Bi_2_S_3_, SnS, boron nitride (BN) such as hexagonal graphene-like (h-BN), and graphene [121,122,123,124,125,126,127,128,129].

An example of PA-ALD used in 2D film deposition is the work of Zhang et al. [129]. They used benzene (C_6_H_6_) and H_2_ plasma at 400 °C as reactants and co-reactant, respectively, in the deposition of graphene on Cu sheets. The authors suggested that the dehydrogenation process of benzene rings formed graphene. The method used was conventional ALD, and non-homogeneous graphene monolayers with grain size < 10 nm were grown below ten cycles. In contrast, ALD cycle values above ten led to high-quality, well-ordered multilayer graphene sheets.

Therefore, ALD 2D materials growth strategies expand the opportunities for 2D materials growth with high controllability, high quality, and large-scale production.

## 4. Why Non-Thermal Plasma on ALE?

The micro and nanofabrication process uses planar technology, i.e., electronic devices are built on the surface of a wafer of semiconductor material. Briefly, during the micro and nanofabrication process, the wafer surface is subjected to several processing steps, in which chemical elements, such as oxygen, and aluminum, are deposited and selectively removed in regions precisely delimited by masks. Each device’s layers have specific electrical characteristics that align with the other layers above and below. The superimposed layers behave like standard electronic components, such as diodes, transistors, capacitors, resistors, and interconnecting conductors. For the manufacture of electronic circuit patterns, it is necessary to use masks with pre-established designs. Pattern transfer is usually accomplished by an etching process that selectively removes unmasked portions of the material. The etching process is carried out through two typical methods: wet etch and dry etch. These two processes generate etching profiles of the isotropic type or anisotropic type, as illustrated in Figure 13. These profiles are differentiated by the level of anisotropy, A, which is calculated by the expression [130]:(1)$$A=1-\frac{E_{h}}{E_{v}}$$
where $E_{h}$ and $E_{v}$ are the horizontal and vertical etching rates, respectively. Figure 13a,b show isotropic etching profiles (A= 0) with lateral under-mask etching and no under-mask etching. Figure 13c shows the partially anisotropic profile ( 0<A<1), and finally, Figure 13d shows the perfectly anisotropic profile (A= 1), i.e., etching on the vertical sidewall [130].

Historically, wet etching was used in microfabrication until the late 1970s. Wet etching uses reagents in liquid form that remove surface layers of material not protected by the mask. As an advantage, wet etching has high selectivity, i.e., the etching rate of a given material M differs from that of material N. As a disadvantage, wet etching is usually of the isotropic type, which is not interesting for specific structures nanometers. In contrast, for structures above ten μm, lateral etching is considered in many cases to be negligible (microelectronic structures). Therefore, it was necessary to develop new etching techniques to reach the nanometric level, resulting in dry etching involving NTP.

Dry or plasma-assisted etching is an anisotropic type (Figure 13d). Through this etching, it is possible to obtain sub-micrometric to nanometric structures in a controlled manner with vertical walls, reducing lateral etching to zero. The great success of applying this plasma etching process induced, in a short time, drastic changes in the micro and nanofabrication processes, with a reduction in the number of process steps, an increase in the efficiency of the devices, and a decrease in production costs of the devices. It would not be possible, for example, the existence of the current processors used in smartphones, tablets, and computers without plasma etching, as this is the only technology capable of corroding anisotropic nanometer lines containing various metallic and dielectric materials. For this reason, plasma plays a dominant role in ALE and is essential in the semiconductor industry. The glow discharge is the basis of plasma-assisted etching because it is in the glow discharge that there is the generation of reactive species, such as atoms, molecules, radicals, and ions generated from molecular gases. Table 3 shows the primary substrates used in plasma-assisted etching, additive gases, and etching atoms [130,131]. It is important to note that some materials are only etched by the ALE technique and are highlighted with a symbol (*) in Table 3.

It is important to note that an ideal etching process is essentially based on chemical mechanisms governed by seven interdependent steps [22,23,130,131]. Three of these steps occur in the gas phase and plasma, and four appear on the etching material’s surface. Table 4 presents the seven steps of the chemical mechanism. However, it is crucial to consider the physical and chemical events involving the etching process in the plasma and etched surface. The fact is that several types of plasma etching can be classified based on three types of phenomenological mechanisms, as outlined in Figure 14a [16,24,130,131]. The first type is a physical etching, also known as sputtering. This mechanism involves the ejection of atoms from the surface due to bombardment by energetic ions, usually from inert gases such as Ar and Xe. Due to the directionality of the mechanism, the film material is removed in a highly anisotropic manner. However, this mechanism is non-selective (all materials are etched at the same rate), causes severe damage to the material surface, and has a low etching rate. The second type is a chemical etching, which involves plasmas that provide reactive atoms or molecules capable of chemically reacting with the material’s surface to form volatile products and cause them to etch. This process presents high selectivity of the material being etched concerning the mask, high etching rates, and low damage to the material surface. However, such a mechanism is typically isotropic, as the neutral reactive species diffuse in all directions, causing an increase in Eh, i.e., etching under the mask. The third type is physico-chemical etching which is based on the synergistic effect of chemically active neutral species and energetic ions incident on the surface.

This type of mechanism offers anisotropic etching controlled by the energy of the incident ions. High etching rates can also be obtained through this mechanism. However, this mechanism may have low selectivity compared to purely chemical etching due to the incidence of high energetic ions. To solve this problem, it is common to use gases that promote the formation of inhibitory precursor molecules, which adsorb or deposit on the substrate, to form protective layers or polymeric films [16]. These inhibitors passivate walls that are not exposed to ion bombardment, promoting better etching anisotropy and adequate selectivity. As we can see in Figure 14b, pressure is a process parameter that controls the type of etching mechanism due to its direct relationship with the energy of ions on the surface. Anisotropy has a similar behavior as it depends on the directionality with which the ions reach the substrate. However, the selectivity reacts oppositely with the decrease in pressure. To achieve the required etching characteristics, it is necessary to vary other process control parameters such as gas composition, electrode geometry, etc. In Figure 14b, we can also see that the reactors that carry out the etching processes are limited to specific pressure values, which limits the applicability of each one to particular methods.

It is important to note that traditional dry corrosion processes depend highly on process pressure, with anisotropy and energy improving at low pressure (Figure 14b). However, when plasma-assisted ALE is analyzed from a process pressure point of view, it becomes one of the least efficient techniques. As seen in Figure 14b, plasma-assisted ALE would have the highest selectivity among dry etch techniques, but anisotropy and energy would be the worst among all established dry etch techniques. However, as ALE is a disruptive technique, its etch mechanism is based on chemistry, temperature, and collisions [45,46]. These three previously mentioned methods are used to overcome the surface binding energy, E_0_, i.e., the chemical process alters the chemical bonds, which decreases E_0_.

In contrast, the thermal process alters the kinetic energy of the vibrations, and finally, the collisions change the bonds due to the kinetic energy of collisions [46,131]. However, the collision process is predominant in plasma-assisted ALE, commonly known as reactive sputtering, which is the primary source of the anisotropic mechanism that affects only atoms close to the impact site.

ALE and plasma-assisted ALE, have simple concepts that focus on dividing the etch into two or more surface reaction steps that can be individually controlled. They are self-limiting, and promote material removal by overcoming the E_0_, but this will only happen if these steps are performed in sequence, i.e., it is necessary to switch the steps. Therefore, ALE is a simple concept, making the physics and chemistry of the process simple, just being based on the construction of combinations of different mechanisms of surface removal by variation of energy (chemistry, temperature, and collision) [45,46,131,132].

## 5. Examples of Applications of PA-ALE in Nanofabrication

The miniaturization of electronic devices and sensors towards the nanoscale has spurred the emergence of atomic scale processes from material deposition to etching, given that the manufacturing of devices approaching the atomic scale requires precise control of all nanofabrication steps [45,133]. ALD and ALE processes have been highlighted as the only thin film techniques for nanofabrication, which can approach the conformality and the aspect ratio independency needed to produce more miniaturized structures [134].

Plasma-assisted atomic layer etching (PA-ALE) has emerged as an outstanding technique to overcome challenges associated with atomic-scale control, material selectivity, etch fidelity, and increasingly complex device architectures, such characteristics are not well addressed by conventional dry etching methods, for example, reactive ion etching (RIE) [135].

The PA-ALE of dielectric materials is a key step in manufacturing nanodevices and nanocomponents of integrated circuits. Currently, self-aligned contacts for advanced transistors are manufactured using fluorocarbon-based PA-ALE processes for SiO_2_ and SiN_x_ [133]. In addition, PA-ALE of SiN-based materials have been also used in the fabrication of device structures such as in the sidewall of transistor gates and the liner-layer at the contact bottom [134].

Another important application of PA-ALE of SiO_2_ is for high-aspect-ratio self-aligned contacts, which are contact hole etching of SiO_2_ in the transistor fabrication formation process [133]. Furthermore, ALE SiO_2_ has been applied to advanced logic devices for the sub-7-nm technology node [134].

An interesting application of PA-ALE was reported by Hsueh et al. [135]. They developed a fabrication using PA-ALE, plasma immersion ion implantation (PIII) and far-infrared laser activation (FIR-LA) to manufacture for the first time, a monolithic 3D SRAM-CIM macro fabricated with BEOL gate-all-around MOSFETs.

PA-ALE with two sequential quasi-self-limiting steps (O_2_ plasma modification step and BCl_3_ plasma removal step) has been explored as a potential alternative to continuous dry etching and digital etching techniques in the fabrication of AlGaN/GaN high-electron mobility transistors (HEMTs) [136].

Recently, the same authors reported the potential of the ALE technique to realize high-performance InAlN/GaN high electron mobility transistors [137]. In this study, different recipes with O_2_ modification times and BCl_3_ removal RF powers were tested for InAlN/AlN/GaN structures. The high efficiency of ALE to precisely control the InAlN etching depth and surface morphology was observed.

Another recent article discussed the ALE of IGZO with CH_4_ using a pulsed plasma and compared it to RIE results [138]. It was observed that the combination of ALE and plasma pulsing enabled controlled reduction of ion-assisted sputtering and redeposition of residues on the patterned IGZO features. The developed process was reproduced on 300 mm wafers, demonstrating suitability in large-scale manufacturing for the intended applications. This is an important finding because IGZO-based thin-film transistors and selector diodes are used in diverse applications such as high-resolution displays, high-density memories, and high-speed computing [138].

A novel transistor type manufactured using ALE was reported by Xiao et al. They developed a vertical C-shaped-channel nanosheet field-effect-transistor (VCNFET) featured precise control of channel-thickness and gate-length by high-quality Si/SiGe epitaxy atomic layer etching with nanometer-scale process control and self-aligned high-k metal gate (HKMG) [139]. The developed process is compatible with mainstream CMOS technology.

Other examples of PA-ALE applications include: (i) ALE of Al_2_O_3_ for possible application in precise etch control and low damage etching of the Al_2_O_3_ layer for use as the interface passivation layer (IPL) between the high-k dielectric and the III–V compound semiconductors [140], (ii) application of ALE for the formation of SiC-based field emitters [141] and (iii) the development of a monolayer MoS_2_ field effect transistor (FET) fabricated after one cycle of ALE, which exhibits electrical characteristics similar to those of a pristine monolayer MoS_2_ FET [142].

In addition to applications in electronic device nanofabrication, the ALE process was also used for nanoscale pattern transfer and high-resolution nanoimprint stamp preparation [143]. Different geometries, loadings, and pitches nanopatterns were fabricated by electron beam lithography on a silicon wafer, whereas ALE was subsequently performed for pattern transfer using a resist as an etch mask. This detailed study allowed us to analyze the different effects and limitations of the process, such as trenching and sidewall tapering. Features as small as 30 nm were successfully transferred into a poly(methyl methacrylate) layer, which demonstrated the great potential of ALE in fabricating nanoimprint stamps with ultrahigh-resolution indicating that it is a very promising technique for further development of nanoimprint lithography (NIL) [143].

## 6. Conclusions

Over the last two decades, NTP has expanded application horizons at low and high pressure (atmospheric pressure). NTP at atmospheric pressure stands out as a valuable tool in agriculture, medicine, biomedicine, dentistry, and areas related to treating and preserving food and beverages [4,5,6,7,8,9,10,11,12,13,14,15,16]. In contrast, low-pressure NTP stands out due to applications in micro/nanoelectronics in various parts of the production chain [17]. With the escalation of ALD tools from the laboratory to the industry, new needs have arisen in the growth of films, 2D films, and surface modifications. However, this technological challenge required a “marriage” with a technique that has been used for decades, cold plasma technology. To show this “marriage,” the present review article sought to establish the potential and challenges of plasma-assisted nanofabrication, emphasizing plasma-assisted atomic layer deposition and plasma-assisted atomic layer etching.

To answer the following question, “Why non-thermal plasma on ALD?” this review presented the primary materials, gases used in plasma generation, and types of reactors [25,32,48,49,50,51,52,53,54,55,56,57,58,59,60,61,62,63]. Devices used to generate plasmas were also presented, such as microwave surfatron, capacitively-coupled plasma, inductively-coupled plasma, hollow-cathode, microwave electron cyclotron resonance, and dielectric discharge [32,48,49,64,65,66,67]. The main applications in nanofabrication were based on oxides, nitrites, sulfides, and phosphates. In recent work, Kim et al. [82] demonstrated how PA-ALD-grown Al_2_O_3_/TiO_2_ nanolaminates could decrease edge shrinkage of flexible OLEDs displays, thus reducing dark spots on monitors and smar TVs, as well as smartphones that are indispensable in the modern world. Another point addressed was the surface modification that PA-ALD causes on the surface of polymers, improving given properties without damaging the base structure. This is due to the low operating temperature of the PA-ALD, which can operate below 80 °C. We have seen that frontier research of knowledge, that is, the production of high-quality 2D materials, joined efforts of the ALD community to develop three distinct methods for the growth of 2D materials (Cai et al. [116]). For example, we saw that Zhang et al. [129], through PA-ALD using benzene (C6H6) and plasma H_2_ at 400 °C as reagents and co-reagents, respectively, grew graphene on Cu sheets.

In an attempt to answer the following question, “Why Non-Thermal Plasma on ALE?” we learned that traditional dry etching processes are heavily dependent on process pressure, with anisotropy and energy improving at low pressure [130]. However, as ALE is a disruptive technique, its attack mechanism is based on chemistry, temperature, and collisions [45,46]. These three previously mentioned methods are used to overcome the surface binding energy, E_0_, i.e., the chemical process changes the chemical bonds, which lowers E_0_. This way, PA-ALE is not dependent on process pressure like conventional etching methods.

In summary, the review provides an expressway for researchers to familiarize themselves with PA-ALD and PA-ALE methodologies and processes. For future studies, we hope to see the application of plasma-activated liquids in ALD, which is little explored but can bring surprising results.

## Figures and Tables

**Figure 1 nanomaterials-12-03497-f001:**
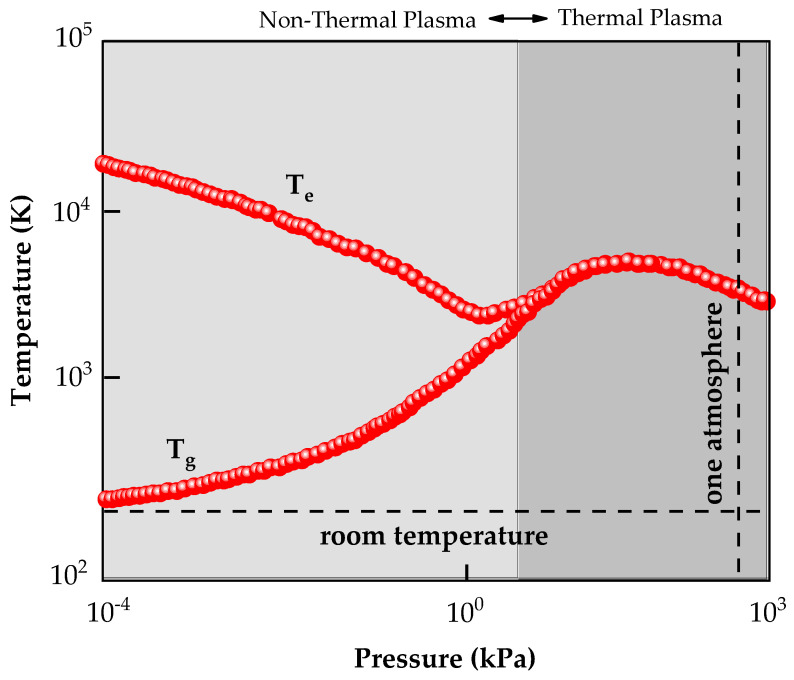
Schematic plot of the electron temperature (T_e_) and the ions and neutral atoms, i.e., gas temperature (T_g_) versus pressure. As can be seen, when the T_g_ is much lower than the T_e_, the plasma is classified as non-thermal plasma. Adapted with permission from Ref. [3]. Copyright 2016, Elsevier.

**Figure 2 nanomaterials-12-03497-f002:**
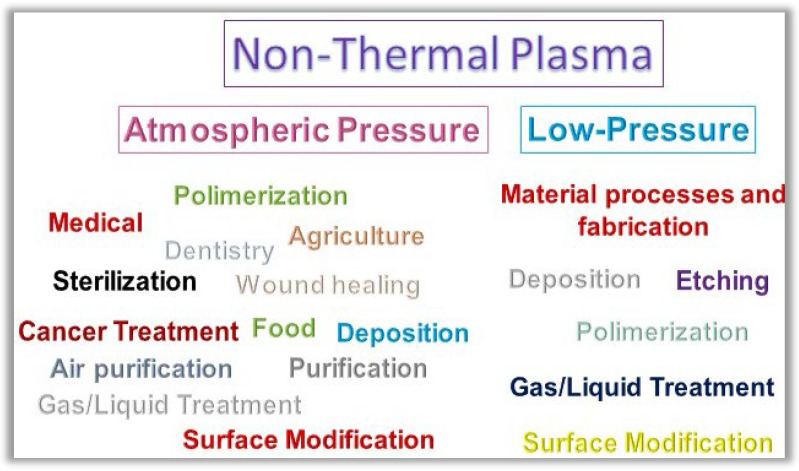
Fundamental applications of the non-thermal plasma at atmospheric and low-pressure.

**Figure 5 nanomaterials-12-03497-f005:**
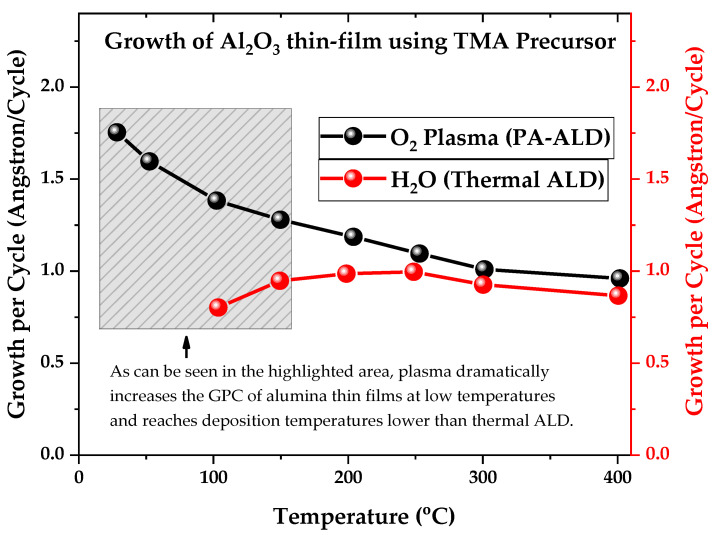
Growth per cycle of Al_2_O_3_ thin films as a function of substrate temperature for both types of ALD reactors, respectively, plasma-assisted atomic layer deposition (PA-ALD) and thermal atomic layer deposition (ALD). Figure adapted with permission from Ref. [57].

**Figure 6 nanomaterials-12-03497-f006:**
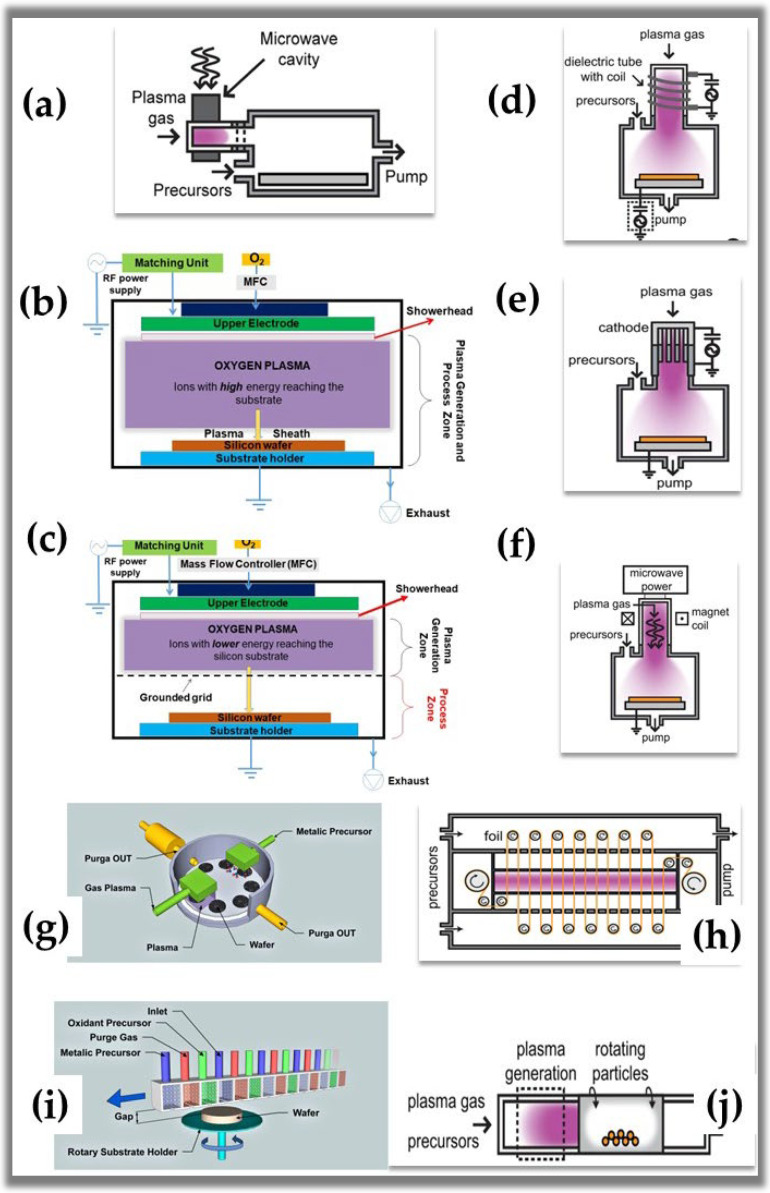
Schematic configuration of reactors used in plasma-assisted ALD [32,48,49,50,64,65,66,67]. Subfigure (**a**) reprinted with permission from Ref. [48]. Copyright 2011, American Vacuum Society. Subfigures (**b**,**c**) reprinted with permission from Ref. [55] by Creative Commons 4.0. Subfigures (**g**,**i**) reprinted with permission from Ref. [32]. Subfigures (**d**–**f**) and (**h**,**j**) reprinted with permission from Ref. [50]. Copyright 2019, American Vacuum Society.

**Figure 7 nanomaterials-12-03497-f007:**
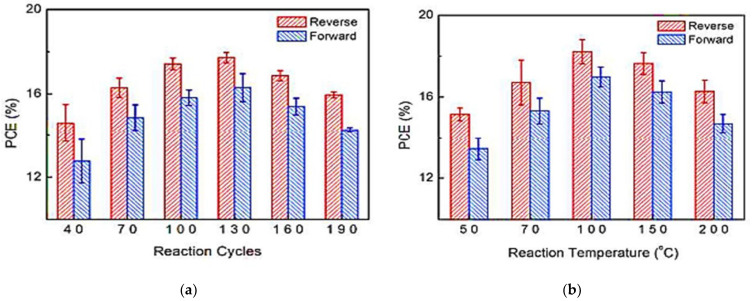
Power conversion efficiency (PCE) as a function of (**a**) reaction cycles; (**b**) process temperature. Reprinted with permission from Ref. [75]. Copyright 2016, Royal Society of Chemistry.

**Figure 8 nanomaterials-12-03497-f008:**
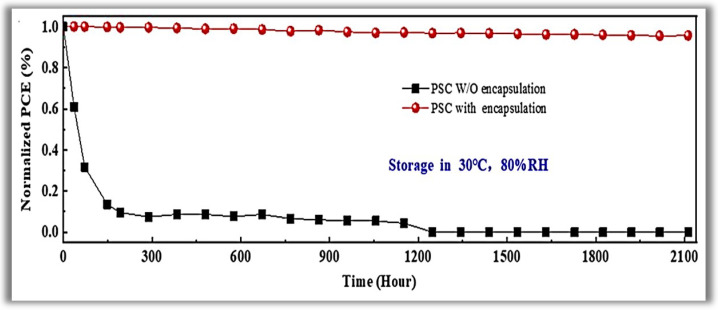
Time evolution of power conversion efficiency for the perovskite solar cells with and without encapsulation under high humidity. Reprinted with permission from Ref. [77]. Copyright 2020, Elsevier.

**Figure 9 nanomaterials-12-03497-f009:**
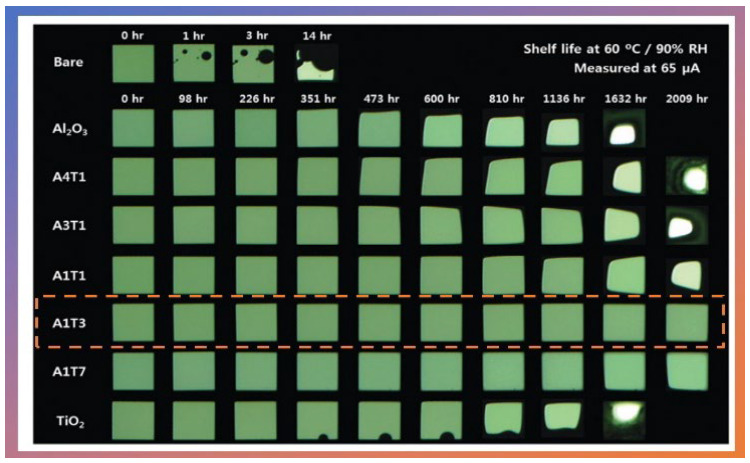
Shelf lives of the OLED devices passivated with and without Al_2_O_3_/TiO_2_ nanolaminates grown via PA-ALD. The A1T3 film highlighted represents the 1:1 atomic ratio of Al_2_O_3_/TiO_2_ with the lowest WVTR value at 60 °C and 90% of RH. Reprinted with permission from Ref. [82]. Copyright 2016, Royal Society of Chemistry.

**Figure 10 nanomaterials-12-03497-f010:**
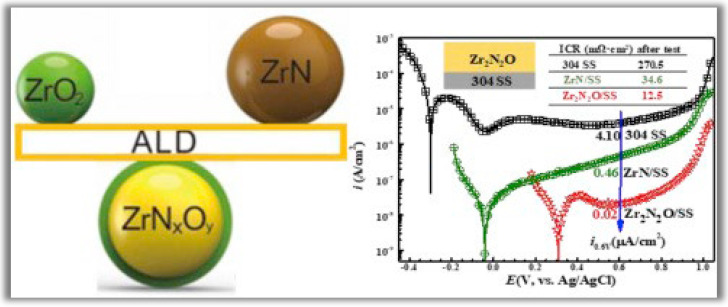
Potentiodynamic polarization curves of 304 stainless steel (SS), ZrN/304 SS, and Zr_2_N_2_O/304 SS. Reprinted with permission from Ref. [93]. Copyright 2018, Elsevier.

**Figure 11 nanomaterials-12-03497-f011:**
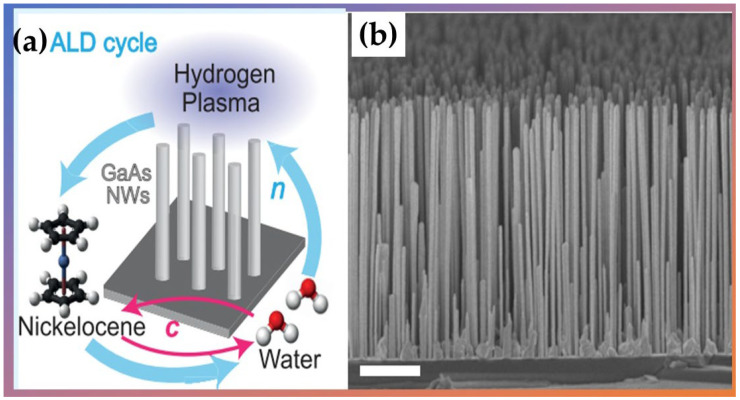
(**a**) Schematic ALD cycle used to synthesize nickel (Ni) nanotubes and (**b**) the scanning electron microscopy (SEM) micrographs of an ensemble of vertical nanowires with PA-ALD-grown Ni shells. Adapted from Ref. [103] with permission by Creative Commons 4.0.

**Figure 12 nanomaterials-12-03497-f012:**
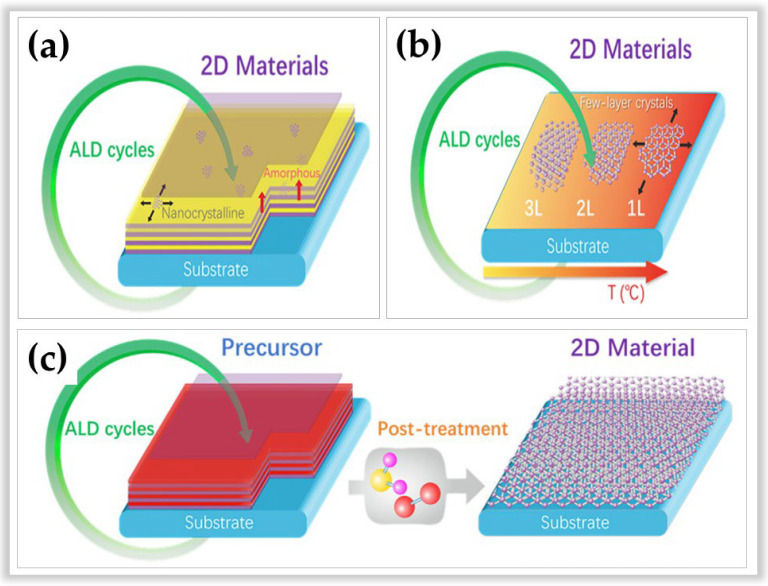
Schematic illustrations of three strategies for growth 2D materials via atomic layer deposition. Subfigure (**a**) represents method 1. Subfigure (**b**) represents method 2. Subfigure (**c**) represents method 3. Reprinted with permission from Ref. [118]. Copyright 2020, Elsevier.

**Figure 13 nanomaterials-12-03497-f013:**
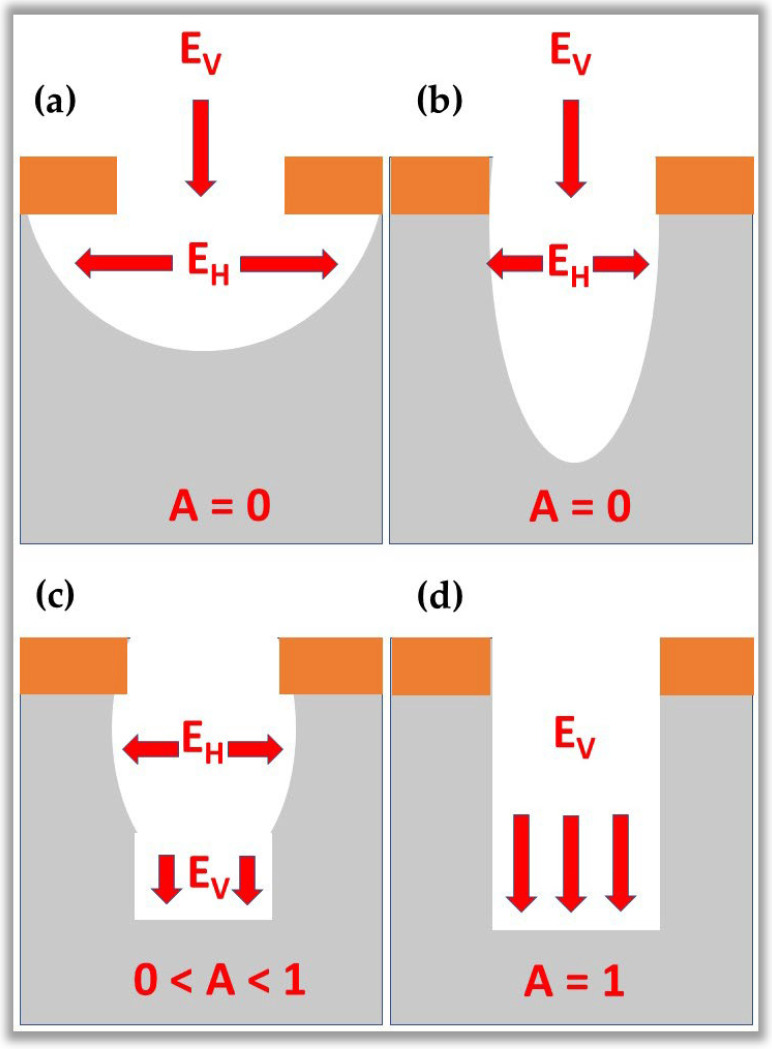
Etching profile types. (**a**) isotropic etching profiles (A=0) with lateral etching under the mask; (**b**) isotropic etching profiles (A=0) without etching under the mask; (**c**) partially anisotropic profile (0<A<1); and (**d**) perfectly anisotropic profile (A=1).

**Figure 14 nanomaterials-12-03497-f014:**
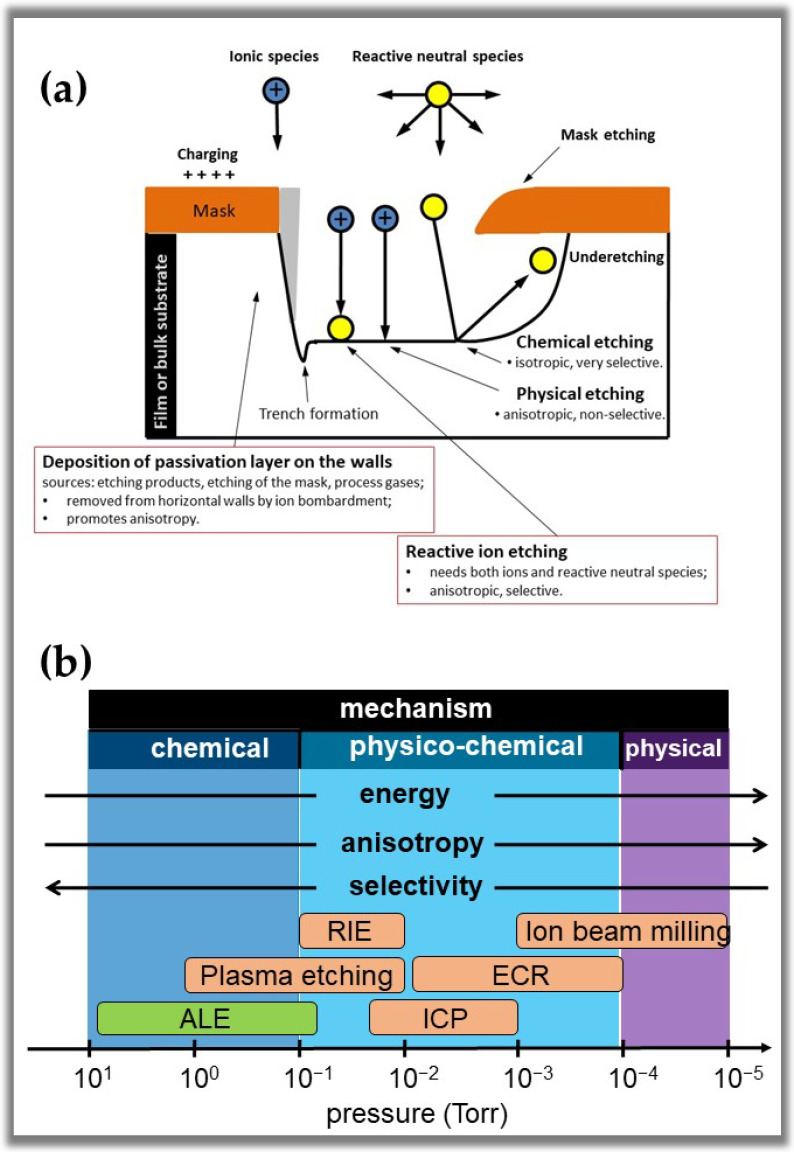
(**a**) Schematic diagram of the main mechanisms governing plasma etching processes and the final shape of the etching profile. (**b**) Dependence on etching mechanisms with the pressure, incident ion energy, anisotropy, selectivity, and reactor types. The direction of the arrow indicates an increase in incident particle energy and anisotropy of the profile formed with decreasing pressure and an increase in selectivity with increasing gas pressure. Symbology: RIE: reactive ion etching, ICP: inductively coupled plasma, ECR: electron cyclotron resonance: ALE: atomic layer etching.

**Table 1 nanomaterials-12-03497-t001:** List of materials grown by PA-ALD.

Types of Materials Grown by PA-ALD	Chemical Elements	Plasma Gases
**Elemental**	Ti, Ta, W, Ru, Co, Ir, Ni, Pd, Pt, Cu, Ag, Au, Al, C, Si, Ge	H_2_
**Oxides**	Li, Mg, Sr, Ba, Ti, Zr, Hf, V, Nb, Ta, Mo, W, Mn, Fe, Ru, Co, Ir, Ni, Pt, Cu, Zn, B, Al, Ga, In, Si, Sn, Bi, La, Ce, Gd, Dy, Er	O_2_, or O_2_/H_2_, or Ar/O_2_
**Nitrides**	Ti, Zr, Hf, V, Nb, Ta, Mo, W, Co, Cu, B, Al, Ga, In, Si, Sn, Gd	N_2_, N_2_/H_2_, NH_3_
**Sulfides**	Mo, W, Zn, Al, Ga	H_2_S
**Fluorides**	Al	SF_6_
**Phosphates**	Ti, V, Fe, Co, Zn, Al	Me_3_PO_4_

**Table 2 nanomaterials-12-03497-t002:** Variety of reactors used on plasma- assisted ALD.

Nomenclature/Plasma Source	Characteristics
Radical-enhanced ALD/Microwave surfatron (Figure 6a)	(1) Only plasma radicals reach the substrate. (2) The film formation reactions are dominated by the chemical behavior of the radicals, i.e., without the participation of energetic ions and electron bombardment in the substrate. (3) Low probability of film decomposition and minor surface damage. (4) The rate of reaction and formation is dominated by the precursors’ reactivity and the desorption of reaction by-products, as in thermal ALD.
Direct mode/Capacitively-coupled plasma (CCP)—without mesh (Figure 6b)	(1) Simple reactor design with proven performance in methods such as plasma-enhanced CVD. (2) Well-established technology in the industry. (3) The substrate is located in the plasma generation zone. (4) The flow of plasma radicals and ions that collide with the deposition surface can have very high energies due to the proximity of the generation source. (5) Uniform deposition over the entire area of the substrate to be exposed to short plasma pulses.
Remote mode/CCP—with mesh (Figure 6c)	(1) Same properties (1) and (2) of the direct plasma without mesh. (2) The substrate is separated from the plasma generation zone by a mesh. (3) The plasma generation zone is separated from the process zone, so damage to the surface ions is significantly suppressed. (4) In contrast with radical-enhanced ALD, electron, and ion densities are not quenched. (5) Uniform deposition over the entire area of the substrate to be exposed to short plasma pulses.
Remote mode/Inductively-coupled plasma (ICP)—with or without bias (Figure 6d)	(1) Due to the remote nature of the remote plasma-enhanced reactor, it is possible to control the composition and properties of the plasma. (2) High degree of flexibility that makes it a suitable candidate for R&D applications. (3) In contrast with radical-enhanced ALD, electron, and ion densities are not quenched.
Remote mode/Hollow-cathode (HC) (Figure 6e)	(1) Same properties of the remote mode—ICP. (2) Hollow cathode plasma source causes minor damage to the plasma than CCP or ICP sources under many conditions. (3) Growth per cycle (GPC (nm/cycle)) is on average 20% higher for oxides and nitrides.
Remote mode/Microwave electron cyclotron resonance (Figure 6f)	(1) Same properties of the remote mode—ICP. (2) Produces a high-density species at low pressure. (3) No risk of contamination of the thin films by sputtered electrode materials, as can occur in CCP.
Carousel mode (Direct)/CCP (Figure 6g)	(1) Same properties of the direct mode—CCP. (2) Multi-wafer reactor. (3) Industrial applications.
Roll-to-roll (Direct)/CCP (Figure 6h)	(1) Ideal for continuous film deposition. (2) Ideal for textiles and flexible substrates.
Atmospheric Spatial DBD (Dielectric barrier discharge) (Figure 6g)	(1) The crucial point of this reactor is the atmospheric pressure deposition.
Low pressure rotary reactor/HC or ICP (Figure 6i)	(1) Ideal for coating particles using plasma ALD on a rotating drum reactor.

**Table 3 nanomaterials-12-03497-t003:** List of chemical materials used in conventional plasma etching and plasma atomic layer etching.

Material Etching	Etching Gas	Additive Gas	Etching Gtoms
Mono and polycrystalline silicon	SF_6_, NF_3_, F_2_, CF_4_, CHF_3_, C_2_F_6_, SiCl_4_, CCl_4_, BCl_3_, CCl_3_F, CCl_2_F_2_, CBrF_3_, HBr	CH_4_, O_2_, H_2_, N_2_, Ar, He	F, Cl, Br
SiO_2_	SF_6_, CH_4_, CHF_3_, C_2_F_6_, C_3_F_8_	CH_4_, O_2_, H_2_, N_2_, Ar	F, F+C
Si_3_N_4_	SF_6_, CH_4_, CHF_3_, C_2_F_6_, C_3_F_8_	CH_4_, O_2_, H_2_, N_2_, Ar	F
Resist	O_2_	-	O
W	SF_6_, CH_4_	O_2_, Ar	F
Al	SiCl_4_, CCl_4_, BCl_3_, Cl_2_	Ar	Cl
* Al_2_O_3_	HF, SF_4_, CHF_3_	-	F
* AlF_3_	HF	-	F
* AlN	HF	-	F
* Co	Cl_2_, O_2_	-	Cl, O
* Cu	O_3_	-	O
* Fe	Cl_2_	-	Cl
* Ga_2_O_3_	HF	-	F
* GaN	XeF_2_	-	F
GaAs	SiCl_4_, CCl_4_, Cl_2_	Ar	Cl
* Ge	Cl_2_	-	Cl
* Graphene	O_2_		O
* Graphite	O_2_	-	O
* HfO_2_	HF, XeF_2_, SF_4_	-	F
* HfZrO_2_	HF, XeF_2_, SF_4_	-	F
* InGaAs	HF	-	F
* InAlAs	HF	-	F
* InGaZnO_4_	HF	-	F
* Mo	O_3_,HF	-	O, F
* Ni	O_2_, HF	-	O, F
* Polymer (Polystyrene)	O_2_	-	O
* TiN	O_3_, CHF_3_/O_2_ downstream plasma	-	F, F+C, O

* Etching by Plasma Atomic layer Etching (PA-ALE).

**Table 4 nanomaterials-12-03497-t004:** Standard etching mechanisms performed by plasma (Dry etching).

Phases	Chemical Mechanism	Site of Chemical Mechanisms
1	Reactive species are generated in the plasma by collisional processes between electrons and neutrals, namely, dissociation, dissociative ionization, and dissociative electron capture, among others;	Gas and plasma
2	these species move from the plasma to the surface of the material to be corroded, by diffusion, in the case of atoms and radicals, and by drift, in the case of ions;	Gas and plasma
3	reaching the surface, these species are adsorbed;	Surface
4	the process of chemisorption of the reactive particles takes place on the surface, i.e., chemical bonds are formed;	Surface
5	these chemical reactions on the surface promote the formation of volatile products;	Surface
6	these volatile products desorb from the surface and	Surface
7	return to the plasma by diffusion, from where they are removed by the pumping system (vacuum pump).	Gas and plasma

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
