# Peer review of "Plasma-Assisted Nanofabrication: The Potential and Challenges in Atomic Layer Deposition and Etching"

_nanomaterials, 2022, doi:10.3390/nano12193497_

Round 1
Reviewer 1 Report
This review paper deals with non-thermal plasma (NTP) technology and its use in atomic layer deposition (ALD) and atomic layer etching (ALE). Over the past two decades, both ALD and ALE have noteworthy expanded their range of application,s especially for the fabrication of nanomaterials.
Hence, the topic is timely and interesting.
The paper is well-written and well-organized. The review covers the field in an exhaustive way.
The manuscript can be published after a minor revision.
Here are a few suggestions for the authors:
1. “ALE is a technique used for the removal of thin layers of material using sequenced and self-limiting reaction steps [xx].” Is there a missing citation here?
2. “Sulfide-based thin films started to stand out, mainly due to the technological appeal of materials such as tungsten disulfide (WS2) and molybdenum disulfide (MoS2), which can revolutionize nanoelectronics.” Add a reference to support this statement (for instance https://doi.org/10.3390/nano10030579).
3. A variety of 2D materials have been reported to date using ALD. The part of section “3. Main Applications of PA-ALD in Nanofabrication” dealing with 2D materials could be extended.
Author Response
REPLY TO REFEREES: NANOMATERIALS - 1946020 entitled “Plasma-assisted nanofabrication: The potential and challenges in atomic layer deposition and etching"
We appreciate the time and effort that the reviewers have dedicated to providing their valuable feedback on our manuscript. We are grateful to the reviewers for their insightful comments on our paper. We have been able to incorporate changes to reflect most of the suggestions provided by the reviewers. We have highlighted the revisions within the manuscript.
Below is a point-by-point response to the reviewers’ comments and concerns.
Reviewer Comments, Author Responses, and Manuscript Changes
Referee(s)' Comments to Author:
Reviewer 1
Comment 1:
1) “The manuscript can be published after a minor revision.
Here are a few suggestions for the authors:
- “ALE is a technique used for the removal of thin layers of material using sequenced and self-limiting reaction steps [xx].” Is there a missing citation here?
Response 1: Thanks for pointing this out. The citation was added on line 128.
Comment 2:
- “Sulfide-based thin films started to stand out, mainly due to the technological appeal of materials such as tungsten disulfide (WS2) and molybdenum disulfide (MoS2), which can revolutionize nanoelectronics.” Add a reference to support this statement (for instance https://doi.org/10.3390/nano10030579).
Response 2: Thank you for pointing that out. The reference was added on line 495.
Comment 3:
- A variety of 2D materials have been reported to date using ALD. The part of section “3. Main Applications of PA-ALD in Nanofabrication” dealing with 2D materials could be extended.”
Response 3: Thanks for pointing this out. To improve the section “3. Main Applications of PA-ALD in Nanofabrication” was added the following text:
Line 246-249: “The third point to be addressed is 2D materials, which play a crucial role in micro/nanoelectronics due belong to a class of advanced materials with odd properties. It is worth mentioning that in Appendix A, there is Table A with a summary of our group's main contributions to the area.”
and the following subsection:
Line 624-663: “3.3 Efforts to produce high-quality 2D layered structures
Since the discovery of graphene, there has been a growing race to produce two-dimensional (2D) layered materials. This continuous search for 2D materials is due to the exceptional properties these materials have. ALD and, consequently, PA-ALD show promise in constructing these materials, mainly due to the unique growth mechanism at the nanometer scale.
In this sense, three distinct methods of ALD in developing 2D materials stand out. Figure 12 illustrates these three methods according to Cai et al. [117]. The first method is based on the conventional ALD process (Figure 12a) and is still the most used method in the processing of amorphous, polycrystalline, or hybrid (amorphous/polycrystalline) 2D materials [118,119]. The net adsorption force between the gaseous reactants and the substrate surface is used. This net adsorption force is always positive with increasing film thickness, allowing for layer-by-layer growth typical of ALD. However, this conventional method allows for slow lateral growth. The second method (Figure 12b) is based on the temperature-focused self-bounding layer synthesis process to define the number of layers [120,121]. Unlike conventional ALD, which is based on the number of cycles to determine the number of layers. However, this method is rarely used and is limited to the growth of MoS2 [120] and WSe2 [121]. The advantages of this method lie in i) the possibility of controlling the layer number precisely, ii) uniformity on a large scale, and iii) the production of 2D materials with a high degree of crystallinity. The disadvantages are that i) the net adsorption on the substrate surface decreases with an increasing number of layers, becoming zero at a saturated number of atomic layers, and ii) the method requires high temperatures (> generally above 500ºC ) to achieve self-limited growth of crystalline 2D materials [117,120,121]. The third method (Figure 12c) is based on two process steps. First, a film is grown at low ALD processing temperatures, then post-treatment is performed at high temperatures to convert the material into 2D. This method has the advantage of manufacturing several 2D materials, such as MoS2, MoSe2, WS2, Bi2S3, SnS, boron nitride (BN) such as hexagonal graphene-like (h-BN), and graphene [122-130].
Figure 12. Schematic illustrations of three strategies for growth 2D materials via atomic layer deposition. Reprinted with permission from Ref. [117]. Copyright 2020, Elsevier.
An example of PA-ALD used in 2D film deposition is the work of Zhang et al. [130]. They used benzene (C6H6) and H2 plasma at 400ºC as reactants and co-reactant, respectively, in the deposition of graphene on Cu sheets. Basically, the authors suggest that the dehydrogenation process of benzene rings formed graphene. The method used was conventional ALD, and it was reported that non-homogeneous graphene monolayers with grain size < 10 nm were grown below ten cycles. In contrast, ALD cycle values above ten led to high-quality, well-ordered multilayer graphene sheets.
Therefore, ALD 2D materials growth strategies expand the opportunities for 2D materials growth with high controllability, high quality, and large-scale production.”
Reviewer 2 Report
The article on plasma-assisted nanofabrication is worth publishing. The literature review is extensive and the authors refer to a large number of works from recent years.
Before publishing, it's a good idea to review group citations and provide more detail. There are minor editorial errors in the article, such as line 125 - no literature is given: [x] or line 344-345 there is not allowed to be transferred to the next line of the power exponent
Author Response
REPLY TO REFEREES: NANOMATERIALS - 1946020 entitled “Plasma-assisted nanofabrication: The potential and challenges in atomic layer deposition and etching"
We appreciate the time and effort that the reviewers have dedicated to providing their valuable feedback on our manuscript. We are grateful to the reviewers for their insightful comments on our paper. We have been able to incorporate changes to reflect most of the suggestions provided by the reviewers. We have highlighted the revisions within the manuscript.
Below is a point-by-point response to the reviewers’ comments and concerns.
Reviewer Comments, Author Responses, and Manuscript Changes
Referee(s)' Comments to Author:
Reviewer 2
Comment 1:
1) “Before publishing, it's a good idea to review group citations and provide more detail.”
Response 1: Thank you for pointing that out. To improve this point, the following text has been added:
Line 246-249: “The third point to be addressed is 2D materials, which play a crucial role in micro/nanoelectronics due belong to a class of advanced materials with odd properties. It is worth mentioning that in Appendix A, there is a summary of our group's main contributions to the area.”
More details is shown in Appendix A.
Comment 2:
2) “ There are minor editorial errors in the article, such as line 125 - no literature is given: [x] or line 344-345 there is not allowed to be transferred to the next line of the power exponente. “
Response 2: Thank you for pointing that out. The errors was fixed.
Reviewer 3 Report
The paper: Plasma-assisted nanofabrication: The potential and challenges in atomic layer deposition and etching represent a very interesting review article in an important field with many possible applications. The terminology used is appropriate and the references is representative for the field. A List of Abbreviations can be used by the authors at the begging of the paper.
The Abstract section can be restructured , at this moment it is more like a part of Introduction. Please move some information in Introduction section and highlight the main findings in the paper field and 1 or 2 perspectives.
Few minor considerations can be considered in order to improve the quality of the paper:
Line 52-55 : Figure 1 requires a reference - or is made by the authors ?
In figure 6 b) si c) - better quality of the images is necessary
Figure 12 is made by the authors ? or necessity of a reference beside the ones mention in text is required
6. Final Remarks section must be better structured , more directions must be highlighted in the field
The authors may consider the work in the field of atomic layer deposition published in Nature journals and mention at reference section.
Author Response
REPLY TO REFEREES: NANOMATERIALS - 1946020 entitled “Plasma-assisted nanofabrication: The potential and challenges in atomic layer deposition and etching"
We appreciate the time and effort that the reviewers have dedicated to providing their valuable feedback on our manuscript. We are grateful to the reviewers for their insightful comments on our paper. We have been able to incorporate changes to reflect most of the suggestions provided by the reviewers. We have highlighted the revisions within the manuscript.
Below is a point-by-point response to the reviewers’ comments and concerns.
Reviewer Comments, Author Responses, and Manuscript Changes
Referee(s)' Comments to Author:
Reviewer 3
Comment 1:
1) “A List of Abbreviations can be used by the authors at the begging of the paper.”
Response 1: Thank you for pointing that out. However, according to the manuscript preparation recommendation (https://www.mdpi.com/journal/nanomaterials/instructions#preparation):
"Acronyms/Abbreviations/Initialisms must be defined the first time they appear in each of the three sections: the abstract; the main text; the first figure or table. parentheses after the written form." However, there is no recommendation for a list of abbreviations.
Comment 2:
2) “The Abstract section can be restructured , at this moment it is more like a part of Introduction. Please move some information in Introduction section and highlight the main findings in the paper field and 1 or 2 perspectives.”
Response 2: Thank you for pointing that out. The Abstract section was restructured.
Line 19-34: “The growing need for increasingly miniaturized devices has placed high importance and demands on nanofabrication technologies with high-quality, low temperatures, and low-cost techniques. In the past few years, the development and recent advances in atomic layer deposition (ALD) processes boosted interest in their use in advanced electronic and nano/microelectromechanical systems (NEMS/MEMS) device manufacturing. In this context, non-thermal plasma (NTP) technology has been highlighted because it allowed the ALD technique to expand its process window and the fabrication of several nanomaterials at reduced temperatures, allowing thermosensitive substrates to be covered with good formability and uniformity. In this review article, we comprehensively describe how the NTP changed the ALD universe and expanded it in device fabrication for different applications. We also present an overview of the efforts and developed strategies to gather the NTP and ALD technologies with the consecutive formation of plasma-assisted ALD (PA-ALD) technique, which has been successfully applied in nanofabrication and surface modification. The advantages and limitations currently faced by this technique are presented and discussed. We conclude this review by showing the atomic layer etching (ALE) technique, another development of NTP and ALD junction that has gained more and more attention by allowing significant advancements in plasma-assisted nanofabrication.”
Comment 3:
3) “Few minor considerations can be considered in order to improve the quality of the paper:
Line 52-55 : Figure 1 requires a reference - or is made by the authors ?
In figure 6 b) si c) - better quality of the images is necessary
Figure 12 is made by the authors ? or necessity of a reference beside the ones mention in text is required.”
Response 3: Thank you for pointing that out. In Figure 1, the reference has been added. The final version of the manuscript had the quality of Figures 6b) and 6c). The former Figure 12 and the current Figure 13 was made by the authors without the need for reference.
Comment 4:
4) “6. Final Remarks section must be better structured , more directions must be highlighted in the field
The authors may consider the work in the field of atomic layer deposition published in Nature journals and mention at reference section.”
Response 3: Thank you for pointing that out. The "6. Final Remarks" section has been restructured as suggested. Regarding publications from Nature journals, some relevant publications were added to the references.
Line 852-891: “6. Final Remarks”
In the last two decades, NTP has expanded application horizons at low and high pressure (atmospheric pressure). NTP at atmospheric pressure stands out as a valuable tool in agriculture, medicine, biomedicine, dentistry, and areas related to treating and preserving food and beverages [4-16]. In contrast, low-pressure NTP stands out due to applications in micro/nanoelectronics in various parts of the production chain [17]. With the escalation of ALD tools from the laboratory to the industry, new needs have arisen in the growth of films, 2D films, and surface modifications. However, this technological challenge required a “marriage” with a technique that has been used for decades, cold plasma technology. To show this “marriage,” the present review article sought to establish the potential and challenges of plasma-assisted nanofabrication, emphasizing plasma-assisted atomic layer deposition and plasma-assisted atomic layer etching.
To answer the following question, “Why non-thermal plasma on ALD?” this review presented the primary materials, gases used in plasma generation, and types of reactors [25,32, 48-63]. Devices used to generate plasmas were also presented, such as microwave surfatron, capacitively-coupled plasma, inductively-coupled plasma, hollow-cathode, microwave electron cyclotron resonance, and dielectric discharge [32,48-49,64-67 ]. The main applications in nanofabrication were based on oxides, nitrites, sulfides, and phosphates. In recent work, Kim et al. [82] demonstrated how PA-ALD-grown Al2O3/TiO2 nanolaminates could decrease edge shrinkage of flexible OLEDs displays, thus reducing dark spots on monitors and smartvs, as well as smartphones that are indispensable in the modern world. Another point addressed was the surface modification that PA-ALD causes on the surface of polymers, improving given properties without damaging the base structure. This is due to the low operating temperature of the PA-ALD, which can operate below 80ºC. We have seen that frontier research of knowledge, that is, the production of high-quality 2D materials, joined efforts of the ALD community to develop three distinct methods for the growth of 2D materials (Cai et al. [117]). For example, we saw that Zhang et al. [130], through PA-ALD using benzene (C6H6) and plasma H2 at 400ºC as reagents and co-reagents, respectively, grew graphene on Cu sheets.
In an attempt to answer the following question, “Why Non-Thermal Plasma on ALE?” we learned that traditional dry etching processes are heavily dependent on process pressure, with anisotropy and energy being improved at low pressure [131]. But as ALE is a disruptive technique, its attack mechanism is based on chemistry, temperature, and collisions [45,46]. These three previously mentioned methods are used to overcome the surface binding energy, E0, i.e., the chemical process changes the chemical bonds, which lowers E0. This way, PA-ALE is not dependent on process pressure like conventional etching methods.
In summary, the review provides an expressway for researchers to familiarize themselves with PA-ALD and PA-ALE methodologies and processes. For future studies, we hope to see the application of plasma-activated liquids in ALD, which is little explored but can bring surprising results.